# Global view on the metabolism of RNA poly(A) tails in yeast *Saccharomyces cerevisiae*

Agnieszka Tudek [1,5 ✉], Paweł S. Krawczyk [1,2,5], Seweryn Mroczek[2,3], Rafał Tomecki[1,3], Matti Turtola [4], Katarzyna Matylla-Kulińska [2,3], Torben Heick Jensen[4] & Andrzej Dziembowski [1,2,3 ✉]

The polyadenosine tail (poly[A]-tail) is a universal modification of eukaryotic messenger RNAs (mRNAs) and non-coding RNAs (ncRNAs). In budding yeast, Pap1-synthesized mRNA poly(A) tails enhance export and translation, whereas Trf4/5-mediated polyadenylation of ncRNAs facilitates degradation by the exosome. Using direct RNA sequencing, we decipher the extent of poly(A) tail dynamics in yeast defective in all relevant exonucleases, deadenylases, and poly(A) polymerases. Predominantly ncRNA poly(A) tails are 20-60 adenosines long. Poly(A) tails of newly transcribed mRNAs are 50 adenosine long on average, with an upper limit of 200. Exonucleolysis by Trf5-assisted nuclear exosome and cytoplasmic deadenylases trim the tails to 40 adenosines on average. Surprisingly, PAN2/3 and CCR4-NOT deadenylase complexes have a large pool of non-overlapping substrates mainly defined by expression level. Finally, we demonstrate that mRNA poly(A) tail length strongly responds to growth conditions, such as heat and nutrient deprivation.

[1] Institute of Biochemistry and Biophysics, Warsaw, Poland. [2] Laboratory of RNA Biology, International Institute of Molecular and Cell Biology, Warsaw, Poland. [3] Institute of Genetics and Biotechnology, Faculty of Biology, University of Warsaw, Warsaw, Poland. [4] Department of Molecular Biology and Genetics, Aarhus University, Aarhus C, Denmark. [5] These authors contributed equally: Agnieszka Tudek, Paweł S. Krawczyk. ✉email: atudek@ibb.waw.pl; adziembowski@iimcb.gov.pl

Polyadenylation of the RNA 3′-end is a ubiquitous modification tied to numerous essential but also conflicting functions. In the budding yeast *Saccharomyces cerevisiae*, the source of polyadenylation defines the fate of the transcript. Pap1, a subunit of the cleavage and polyadenylation factor (CPF) complex, polyadenylates newly produced mRNAs to promote their stability and export to the cytoplasm. In contrast, Trf4 or Trf5 poly(A) polymerases constituting subunits of paralogous nuclear TRAMP4 or TRAMP5 (Mtr4-Air1/2-Trf4/5) complexes facilitate degradation or processing of nuclear noncoding RNAs (ncRNAs) by the exosome complex.

TRAMP4/5 complexes can act co- or post-transcriptionally. The Nrd1-Nab3-Sen1 (NNS) system mediates transcription termination at cryptic unstable transcripts (CUTs) and small nuclear/nucleolar RNA (sn/snoRNA) loci and physically recruits TRAMP4 together with the exosome complex to those RNAs[1,2]. Independent of transcription, TRAMP5 participates in the control of mRNA stability and pre-ribosomal RNA (rRNA) processing[3–5]. Concerted polymerase and Mtr4-mediated helicase activities of TRAMP4/5 promote the 3′–5′ exonucleolytic activities of the exosome Rrp6 and Dis3 subunits. However, polyadenylation is not absolutely required for exonucleolysis as a catalytically dead Trf4 partially rescues the *trf4Δ* growth and decay phenotypes[6]. The length of in vivo TRAMP-dependent poly(A) tails is debated. Selected studies in vitro show that the Mtr4 RNA helicase limits Trf4/5 polyadenylation to 3–5 adenosines[7,8]. Indeed in vivo, high-throughput RNA-sequencing (RNA-seq) experiments detected exclusively such oligoadenylated terminal or internal decay intermediates[3,9], which, however, might not be representative because of the detection limit of the sequencing method. In contrast, other in vitro studies have shown that TRAMP4/5 can produce poly(A) stretches many times longer[6,10].

The evolutionarily conserved CPF complex mediates transcription termination at protein-coding genes coupling endonucleolytic cleavage to Pap1-mediated polyadenylation of the transcript. The bulk of yeast poly(A) tails is a maximum of 70 adenosines (As) long. However, in vitro studies showed that Pap1 can produce much longer poly(A) tails[11–13], raising the question of the in vivo polyadenylation limit upon mRNA de novo synthesis.

A poly(A) tail of at least 48 As is required for nuclear mRNA stability and efficient export facilitated by the nuclear poly(A) binding protein Nab2[14–16]. Nab2 is a part of the assembled co- and post-transcriptionally export-competent messenger ribonucleoprotein (mRNP) complex, which contains the main export-adapter, Mex67/Mtr2 dimer and supporting THO/TREX complex (Thp2, Tho2, Tex1, Hpr1, Mft1)[17]. Nab2 is later replaced in the cytoplasm by Pab1, which brings together the poly(A) tail with the cap-bound translation initiation factors[18]. In the cytoplasm, the mRNA poly(A) tail can be targeted by two deadenylase complexes, PAN2/3 and CCR4-NOT[19]. In vitro, PAN2/3 action is enhanced by binding to multimerized Pab1[20,21], while CCR4-NOT preferentially degrades mRNAs with no or low Pab1 load[22]. Those data might support the model postulating that PAN2/3 mediates the first rapid step of deadenylation, followed by the action of the CCR4-NOT complex resulting in mRNA decapping and 5′–3′ decay by Xrn1[19,20,23,24]. However, comprehensive in vivo transcriptomic evidence in yeast supporting this model is still lacking.

Here, we defined the contribution of all *S. cerevisiae* poly(A) polymerases, the exosome together with its cofactors, and deadenylase complexes to shaping poly(A) tail length and abundance of all RNA biotypes. We used the nanopore direct RNA sequencing (DRS) technology[25] to study the poly(A)+ RNA fraction, which improves the estimation of long polyadenosine reads compared with poly(A) test (PAT)-based methods[26] or TAIL-seq[27]. We found that CUT, rRNA, and snRNA 3′ ends were mainly polyadenylated by TRAMP4/5 and in the exosome-deficient background had poly(A) tails comparable in length to mRNAs. De novo-produced mRNA poly(A) tails were 50 As long, with an upper limit of 200 As. At steady state, mRNA poly(A) tails were deadenylated to 40 adenosines on average. Surprisingly the nuclear exosome, assisted by TRAMP5, extensively trimmed mRNA poly(A) tails, while TRAMP4 adenylated mRNAs in exosome- and Pap1-deficient cells. We found that PAN2/3 and CCR4-NOT both mediate the first cytoplasmic deadenylation phase of mRNAs. Each complex has its dedicated set of substrates best defined by expression level, with PAN2/3 targeting abundant mRNAs and CCR4-NOT being more specialized.

## Results

### DRS efficiently estimates mRNA abundance and poly(A) tail length of yeast RNA.

We determined the range of poly(A) tail lengths of yeast RNAs and their abundance, using nanopore direct RNA sequencing (DRS) of the poly(A)+ RNA fraction extracted with oligo-(dT)$_{25}$ beads. Such RNA, efficiently depleted of non-polyadenylated ncRNA species (Supplemental Fig. 1a), was a substrate for library preparation based on splint ligation with (dT)$_{10}$ and reverse transcription to produce an RNA:cDNA hybrid (Fig. 1a)[25]. DRS counts aligned well with previously published data derived from RNA-seq on poly(A)+ fraction or total RNA after ribodepletion, as well as tiling arrays[28,29] (Supplemental Fig. 1b), indicating that quantitatively DRS is comparable to other genome-wide techniques. Moreover, DRS counts strongly correlated with one another, whereas, as previously shown, poly(A) tail lengths had a higher degree of variability[30] (Supplemental Fig. 1c–e).

DRS protocol strongly recommends poly(A)+ RNA selection using oligo-(dT)$_{25}$ beads to both obtain sufficient amounts of RNA (50–200 ng) for library synthesis and deplete the sample from abundant ncRNAs (rRNA), which impede sequencing due to strong secondary structures blocking the nanopores. However, such an approach can discriminate against non- or oligoadenylated transcripts, which we sought to quantify. Analysis of previously published short-read RNA-seq datasets[31,32] indicated that the non-adenylated mRNAs fraction is slim as in vitro polyadenylation prior to (dT)-priming led to a strong stabilization of sn-/snoRNA and other poly(A)⁻ ncRNAs, but had little effect on the levels of mRNAs (Supplemental Fig. 1f). To approximate the levels of oligoadenylated mRNAs, that could be lost during the poly(A)+ enrichment, we subjected a total RNA sample to DRS, which is equivalent to (dT)-priming for RNA-seq. This was possible as analysis of artificial spike-ins indicated that the DRS chemistry and data analysis are compatible with oligoadenylated transcripts (Supplemental Fig. 1g). Poly(A)+ and total DRS datasets strongly correlated with one another in terms of mRNA abundance and poly(A) tail length (Fig. 1b), but the total RNA dataset was enriched in reads bearing tails mostly shorter than 10 As (Supplemental Fig. 1h). We found that highly expressed transcripts were especially enriched with those short-tailed RNAs leading to the decrease in mean poly(A) tail length estimation and a slight increase in abundance compared to poly(A)+ DRS set (Supplemental Fig. 1i), which was cross-compared and validated by reverse-transcription qPCR (Supplemental Fig. 1j). Collectively, this indicates that each mRNA has an individual fraction of oligoadenylated species. However, we only found 840 mRNAs with a short-tailed fraction large enough to strongly affect the mean poly(A) tail and abundance estimations (Supplementary Data 1). Consequently, we worked with the poly

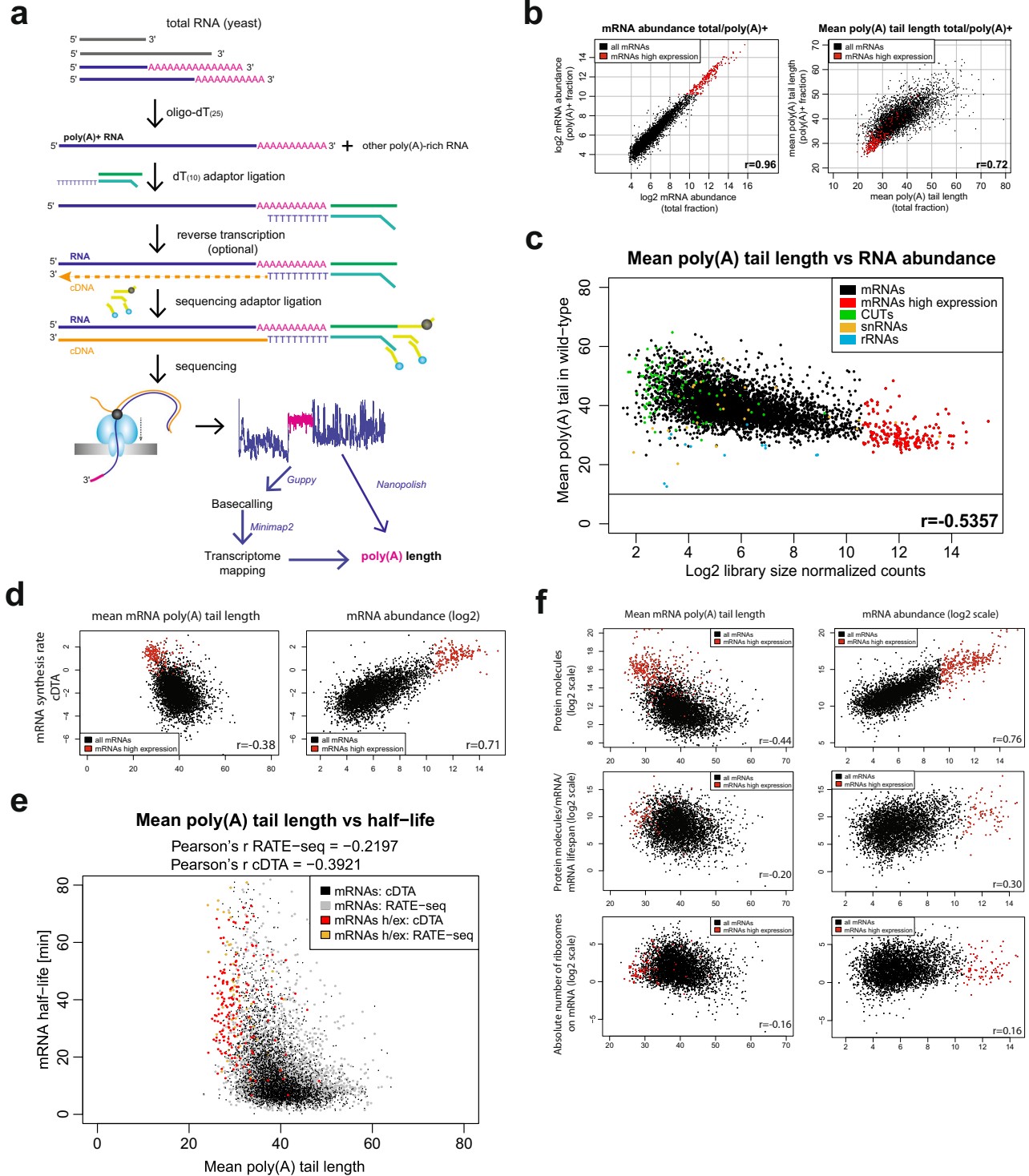

(A)$^+$ fraction for further analysis as it proved to be a good approximation for the total sample.

**mRNA poly(A) tail lengths anticorrelate with transcript abundance and are not linked with transcription or translation rates.** Under steady-state 30 °C or 25 °C growth conditions, poly (A)$^+$ RNA isolated from WT BY4741 and W303 yeast strains had mean and median poly(A) tails of 40 and 37 As, respectively. However, the poly(A) tail length distribution varied significantly between individual transcripts (Fig. 1c). Highly expressed mRNAs had relatively short tails with a mean of 30 As and a median of 26 As, reflecting similar tendencies in other organisms[30,33]. In contrast, poly(A) tail lengths of all transcripts, both mRNAs and ncRNAs, with lower expression varied from 30 to 60 As. Such heterogeneity of poly(A) tail lengths increased gradually with decreasing mRNA abundance.

To pinpoint the factor best defining the mean mRNA poly(A) tail length at steady state, we cross-compared our DRS datasets to published data describing key aspects of transcript biogenesis and function. mRNA synthesis rate estimates best defined DRS mRNA abundance but poorly correlated with mean poly(A) tail lengths[34–36] (Fig. 1d). At steady state, the mean poly(A) tail did

**Fig. 1 RNA poly(A) tail length anticorrelates with RNA expression. a** Schematic diagram of the DRS technique used to estimate poly(A) tail lengths. The poly(A)$^+$ fraction is isolated from total RNA; then libraries are prepared according to the protocol, with the first step of (dT)$_{10}$ adapter ligation. After sequencing bioinformatic analysis steps are performed: basecalling, mapping to transcriptome, and estimation of the length of the poly(A) tail (marked in pink) by Nanopolish using a raw signal readout. **b** Scatterplots showing in the left panel the comparison of log2 mRNA abundance in DRS libraries prepared using the poly(A)$^+$ fraction or total sample and in the right panel the mean poly(A) tail length estimates in both library types. **c** Relationship between RNA abundance and the mean poly(A) tail length of different RNA biotypes, averaged from three independent WT control datasets (two from W303 and one from BY genetic backgrounds, each composed of a minimum of two biological repeats). The cutoff line shows the limit of the efficient DRS poly(A) tail estimation method (10 A). **d** Relationship between DRS-defined mean poly(A) tail length (left panel) and log2-scaled mRNA abundance (right panel) relative to log2-scaled mRNA synthesis rate from cDTA dataset[36]. **e** Relationship between mean mRNA poly(A) tail length and transcripts half-life from cDTA or RATE-seq datasets[34,36]. "h/ex" in the key designates highly expressed mRNAs, which are marked in red (for cDTA) or orange (for RATE-seq). **f** Relationship between mean poly(A) tail length (left panels), log2-scaled mRNA abundance (right panels), the log2-scaled number of protein molecules per cell[37] (upper panels), the log2-scaled number of protein molecules that are produced from a given transcript during its lifespan (middle panels) and log2-scaled raw ribosome occupancy estimation (bottom panels; factor b and w from Siwiak and Zielenkiewicz[38]). Pearson's correlation coefficients are indicated on each scatterplot.

not correlate with mRNA half-lives either, although some mRNAs with short poly(A) tails, including highly expressed ones, were more abundant in the long half-life mRNAs group[34,36] (Fig. 1e). The number of protein molecules per cell correlated strongly with mRNA abundance and anticorrelated with mean poly(A) tail length[37] (Fig. 1f, top row). The latter could be a sign of co-translational mRNA deadenylation. However, when considering translational efficiency, this interdependence was virtually lost[38] (Fig. 1f, middle and bottom rows). Given those results, we hypothesize that mRNA poly(A) tail lengths are likely to be defined by exonucleolysis, at least in part independently of translation.

**Pap1 produces poly(A) tails of mostly 50 As with the limit of 200 As, and Trf4 can partially compensate for its depletion**. To define the extent of exonucleolysis, we first sought to determine the length of de novo produced poly(A) tails. Poly(A) tail length at steady state is a function of nuclear polyadenylation and presumably mostly cytoplasmic deadenylation. Therefore, to minimize the impact of deadenylation on poly(A) tail length of newly made mRNAs, we perturbed nuclear export. As previously shown in *mft1Δ* cells, export is impaired at high temperatures, whereas in cells depleted for Mex67, using the anchor-away (AA) system, export is entirely blocked[32,39,40] (Fig. 2a). We rapidly shifted cells from 25 to 38 °C for 12–15 min, allowing de novo production of a set of heat-stress mRNAs, which in both export-blocked strains had poly(A) tails 50 As long on average (47 A and 51 A for *mft1Δ* and Mex67-AA, respectively, Fig. 2a and Supplemental Fig. 2a). The maximum upper limit of mRNA poly(A) tail lengths was roughly 200 As in Mex67-depleted cells and 163 in *mft1Δ*, compared to 140 in both controls (Fig. 2a, b and Supplemental Fig. 2a). Note that in both growth conditions (rich or minimal media used for *mft1Δ* and Mex67-AA, respectively), de novo Pap1 produced poly(A) tails were similar in length despite a general steady-state shortening of poly(A) tail lengths in minimal media (see also the last section).

To evaluate the role of the nuclear poly(A) polymerase Trf4 in the biogenesis of mRNA poly(A) tails, we inhibited the Pap1 activity using a thermosensitive *pap1-1* mutant[41]. As previously shown, deletion of the nuclear exosome subunit *RRP6* in *pap1-1* partially suppressed the 28.5 °C thermosensitivity of *pap1-1* cells (Fig. 2c). It also restored the levels of six arbitrarily tested transcripts to those observed in a WT strain (Supplemental Fig. 2b). We hypothesized that partial exosome inactivation in *pap1-1* set the space for non-canonical poly(A) polymerases to adenylate mRNAs and restore their functionality. Indeed, the loss of *TRF4* exacerbated the growth defect of *pap1-1* cells and did not support mRNA stability[41] (Fig. 2c and Supplemental Fig. 2b). We thus sequenced poly(A)$^+$ RNA from relevant strains and used a

reverse-transcription quantitative polymerase chain reaction (qPCR)-derived coefficient (RT-qPCR) to normalize the DRS datasets. This was required because we noticed that a marked decrease in poly(A)$^+$ RNA yield from *pap1-1* and *pap1-1 trf4Δ* cells resulted in the loss of a difference in mRNA abundances compared to WT cells when only library size normalization was employed (Supplemental Fig. 2c, d). In the *pap1-1* mutant, mRNA poly(A) tail shortening was proportional to the poly(A) tail length in control cells (Supplemental Fig. 2e), suggesting that the mutation impairs polymerase processivity. mRNA poly(A) tails were significantly shortened by 6.86, 3.73, and 7.66 As on average in the *pap1-1*, *pap1-1 rrp6Δ*, and *pap1-1 trf4Δ* backgrounds, respectively (Fig. 2d, blue series). Thus, loss of *RRP6* led to a partial rescue of growth, mRNA abundance, and poly(A) tail length in *pap1-1* background. Since no such effect was seen in *trf4Δ*, we conclude that the mechanism of rescue is likely mostly related to Trf4/5-mediated polyadenylation, which is only visible in exosome-deficient backgrounds[42].

Previous studies suggested a link between deficient adenylation and decay of the transcript[41]. Though this was confirmed in our analysis by the absolute level of poly(A)$^+$ RNA, especially for highly expressed mRNAs, we found no global correlation between absolute changes in mean poly(A) tail length and fold changes in RNA abundance for the *pap1-1*, *pap1-1 rrp6Δ*, and *pap1-1 trf4Δ* backgrounds (Fig. 2e–g, for single-gene examples see Fig. 2h, and Supplemental Fig. 2f). The lack of such correlation indicates that some mRNAs are adenylated correctly in *pap1-1*, and the steady-state mRNA abundance/poly(A) tail length pattern results from other nuclear or cytoplasmic compensatory mechanisms.

**ncRNAs possess long poly(A) tails, and TRAMP5 assists Rrp6 in mRNA poly(A) tail trimming**. To evaluate the direct impact of the nuclear exosome and TRAMP4/5 on poly(A) tail biogenesis and the stability of different RNA biotypes, we produced DRS datasets for WT, *trf4Δ*, *trf5Δ*, and *rrp6Δ* strains grown at 30 °C.

We first inspected nuclear ncRNAs, which are the predominant exosome targets. As expected, the sum of reads mapping to CUT, sn-/snoRNA, and rRNA loci was significantly increased in the *rrp6Δ* dataset compared with WT (Fig. 3a, bottom panel). Increased adenylation of usually non-polyadenylated ncRNAs, such as 25S rRNA, *SCR1*, and *LSR1*, was previously reported[28,43] and validated by RT-qPCR (Supplemental Fig. 3a). Importantly, fold stabilization of CUTs in our DRS datasets was comparable to previous reports[28] (Supplemental Fig. 3b), suggesting that despite the oligo-(dT) enrichment, we detected the bulk of those ncRNAs. Even though Trf4/5 are expected to oligoadenylate ncRNAs[7,8], to our surprise, poly(A) tails of CUTs, sn-/snoRNAs, and rRNAs were on average 45, 54, and 52 As long in *rrp6Δ* cells. Such long tails were also detected in the WT strain (Fig. 3a, top

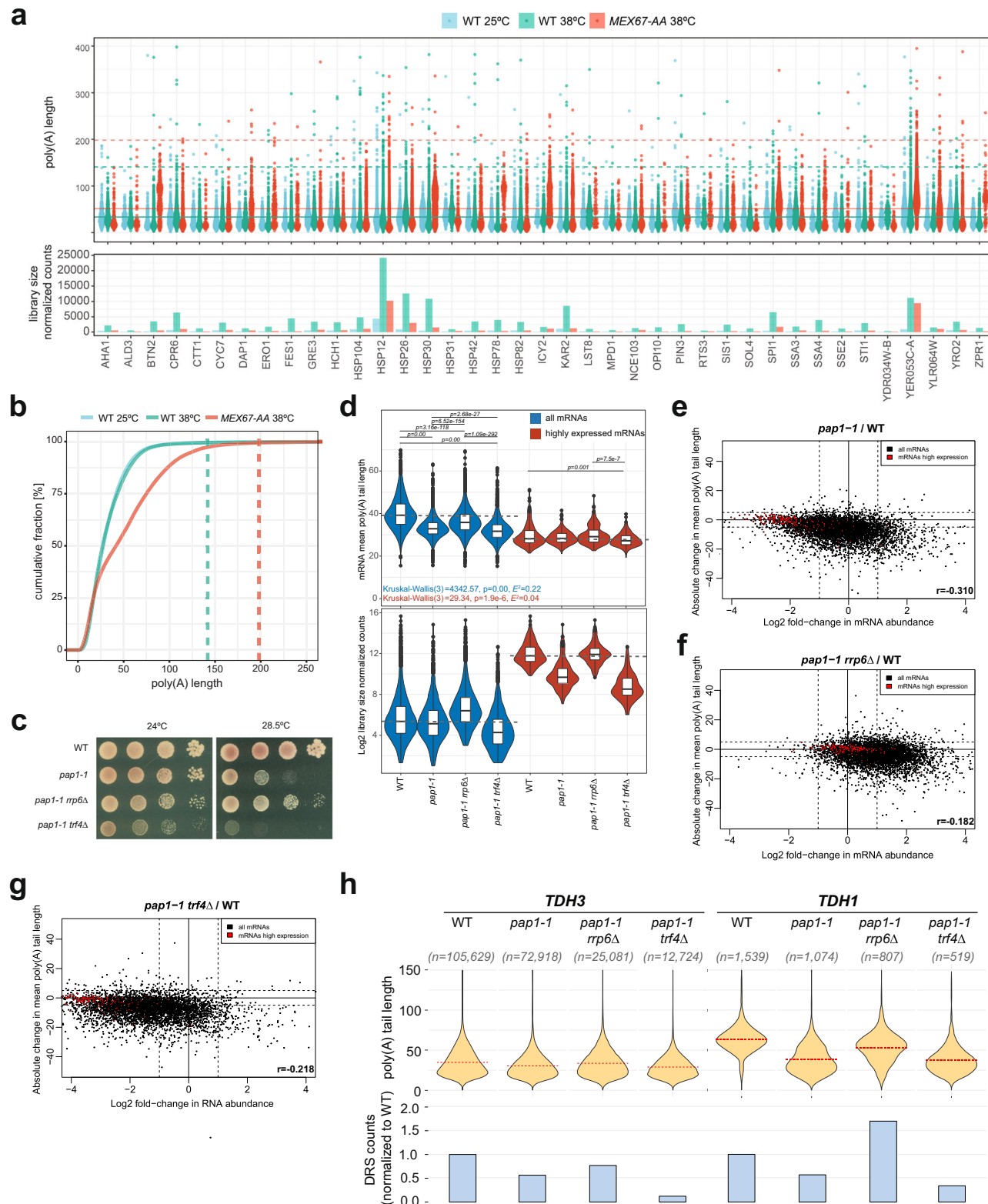

panel). Observed ncRNA adenylation was partially dependent on Trf4, manifested by both mean poly(A) tail length and abundance in the *trf4Δ* and *rrp6Δ trf4Δ* datasets[6] (Fig. 3a). We validated the abundance and adenylation pattern of *snR13* in the poly(A)$^+$ fraction by Northern blot analysis (Supplemental Fig. 3c, compare left to right panel). Consistent with the role of TRAMP5 in rRNA maturation[5], we observed a decrease in the abundance and poly(A) tail length of many rRNA precursors in *trf5Δ* cells (Fig. 3a).

This recapitulated the known functions of the exosome and TRAMP4 in ncRNA degradation and strongly suggested that TRAMP4/5 produce poly(A) tails comparable in length to mRNAs.

The deletion of *RRP6* or its cofactors *TRF4* or *TRF5* did not affect the abundance of protein-coding genes. However, poly(A) tails were elongated by an average of 3.4 and 4 As in *rrp6Δ* and *trf5Δ* cells, respectively, remaining on average unaltered in *trf4Δ*

**Fig. 2 Pap1 produces poly(A) tails of mostly 50 adenosines with the limit of 200 adenosines, and Trf4 can partially compensate for its depletion.**
**a** The top panel shows a beeswarm plot for poly(A) tail length distribution of a collection of heat-induced transcripts in control and Mex67-depleted cells. Means and 99.5% quantile are marked with solid and dashed lines, respectively, for Mex67-AA with or without rapamycin at 38 °C. The bottom panel shows library size normalized counts for those mRNAs. **b** Empirical cumulative density function plot for poly(A) tail lengths of heat-induced transcripts in control (Mex67-AA -rapamycin: WT 38 °C) and Mex67-depleted cells (+rapamycin). Dashed lines represent 99.5% quantile for control 38 °C (green) and Mex67-AA 38 °C (red). **c** Growth of WT, *pap1-1*, *pap1-1 rrp6Δ*, and *pap1-1 trf4Δ* strains. Equal numbers of cells of each strain were spotted on YPDA plates in tenfold serial dilutions and grown for 3 days at 24 or 28.5 °C. **d** Violin plot of mean mRNA poly(A) tail lengths (top) and mRNA abundance (bottom) of all (blue series) and highly expressed mRNAs (red series n = 184) for WT (n = 5079), *pap1-1* (n = 5161), *pap1-1 rrp6Δ* (n = 5054), and *pap1-1 trf4Δ* (n = 5770) strains. For the upper panel, Kruskal–Wallis test *P* value and Holm-corrected *P* values for two-sided Dunn pairwise test are shown. Dashed gray lines show the mean values for WT strain to ease visual comparison. Box plots show median value (solid bold line) and 1st and 3rd quartiles, whiskers represent 1.5 × IQR (interquartile range). Outliers are marked as black points. **e–g** Scatterplots of the relationship between log2 fold change in mRNA abundance (*y* axis) and absolute change in mean mRNA poly(A) tail for *pap1-1* (**e**), *pap1-1 rrp6Δ* (**f**), and *pap1-1 trf4Δ* (**g**) cells (*x* axis), compared with WT. Dashed lines mark a twofold change in expression level and a 5-adenosine change of mean poly(A) tail length. **h** Poly(A) tail length distribution of *TDH3* and *TDH1* in *pap1-1* mutant cells (upper panel) and relative abundance changes for each condition (bottom). The number of DRS reads for each condition is shown for each transcript. Red dashed lines show the mean for each condition. Pearson's correlation coefficients are indicated on each scatterplot.

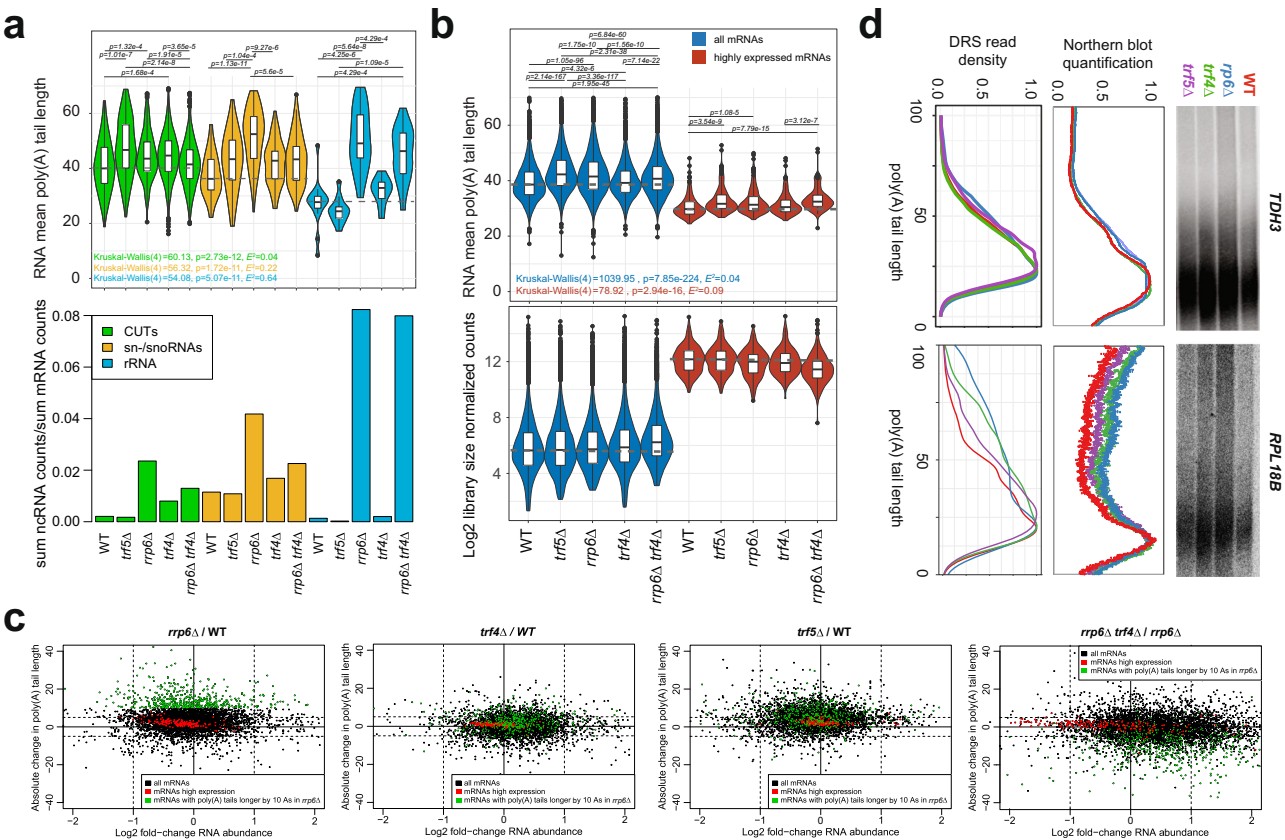

**Fig. 3 mRNA poly(A) tail trimming by the exosome is stimulated by Trf5. a** The top panel shows a violin plot of mean poly(A) tail lengths of nuclear ncRNAs detected in WT and exosome mutant cells. The number of RNAs detected n = [WT, *trf5Δ*, *rrp6Δ*, *trf4Δ*, *rrp6Δ trf4Δ*] was for CUTs = [240, 129, 519, 421]; sn/snoRNAs = [47, 40, 63, 53, 56] and rRNAs = [15, 12, 23, 14, 23]. The bottom panel shows the sum of all DRS counts that map to those ncRNA loci, normalized to the sum of counts that were mapped to protein-coding loci. For the upper panel, Kruskal–Wallis test *P* value and Holm-corrected *P* values for two-sided Dunn pairwise test are shown. Dashed gray lines show the mean values for WT strain to ease visual comparison. Box plots show median value (solid bold line) and 1st and 3nd quartiles, whiskers represent 1.5 × IQR (interquartile range). Outliers are marked as black points. **b** Violin plot of the mean mRNA poly(A) tail length distribution and log2-scaled mRNA abundance in the top and bottom panels, respectively, for WT (n = 5269), *trf5Δ* (n = 5009), *rrp6Δ* (n = 4951), *trf4Δ* (n = 5258), and *rrp6Δ trf4Δ* (n = 4928) strains (highly expressed mRNAs n = 184). For the upper panel, Kruskal–Wallis test *P* value and Holm-corrected *P* values for two-sided Dunn pairwise test are shown. Dashed gray lines show the mean values for WT strain to ease visual comparison. Box plots show median value (solid bold line) and 1st and 3nd quartiles, whiskers represent 1.5 × IQR (interquartile range). Outliers are marked as black points. **c** Relationship between absolute change in mean mRNA poly(A) tail length and fold change in mRNA abundance for (from left to right) *rrp6Δ*, trf4Δ, and *trf5Δ* compared with WT and (rightmost) *rrp6Δ trf4Δ* cells compared with *rrp6Δ*. mRNAs with poly(A) tails that were longer by at least ten adenosines in *rrp6Δ* are highlighted in green in the *rrp6Δ* and *rrp6Δ trf4Δ* plots. Dashed lines mark a twofold change in expression and 5-adenosine change of mean poly(A) tail length. Pearson's correlation coefficients are indicated. **d** The rightmost panel shows the Northern blot analysis of *TDH3* and *RPL18B* mRNA 3′-UTRs that were cleaved off via RNaseH digestion and run on a high-resolution acrylamide gel for WT, *rrp6Δ*, *trf4Δ*, and *trf5Δ* cells. The middle panel shows the Northern signal distribution, averaged from two biological replicates. The leftmost panel shows the poly(A) tail length distribution in the DRS datasets.

cells (mean change = 0.5 adenosines; Fig. 3b, c, Supplemental Fig. 3d, e). Most affected mRNAs were expressed at low or moderate levels and had medium-long poly(A) tail in WT cells (Supplemental Fig. 3f, g). This suggests a role of TRAMP5 in mRNA poly(A) tail trimming by the exosome. However, the relatively modest correlation of poly(A) tail changes in *rrp6Δ* and *trf5Δ* cells (Supplemental Fig. 3e) might indicate that TRAMP5 can also act as a cofactor of the other exosome nuclease Dis3. Based on the total RNA fraction and Northern blot analysis, we validated changes in the poly(A) tail length of *RPL18B* mRNA, which had elongated poly(A) tails in all deletion strains, and *TDH3* mRNA, which remained largely unaffected (Fig. 3d).

Finally, we investigated whether some mRNA poly(A) tail elongation in *rrp6Δ* cells can be attributed to TRAMP4-mediated adenylation assessing poly(A) tail length in a double *rrp6Δ trf4Δ* strain compared with *rrp6Δ* (Fig. 3c and Supplemental Fig. 3h–j). Many mRNAs, which had poly(A) tails that were at least 10 As longer in the *rrp6Δ* strain (green dots), tended to be shortened in the double *rrp6Δ trf4Δ* strain, indicating that TRAMP4 indeed polyadenylated them. However, poly(A) tails of mRNAs were still globally elongated in *rrp6Δ trf4Δ* strain by an average of 2.3 As compared to WT (Fig. 3b). This suggests that poly(A) tail elongation in *rrp6Δ* is not only related to activation of Trf4-polymerase activity.

Overall, these results indicate that the length of poly(A) tails of some mRNAs is regulated by Rrp6, mainly resulting from the TRAMP5-enhanced exonucleolytic trimming of poly(A) tails. However, adenylation of a few mRNAs may be performed by TRAMP4.

**Dis3 and Rrp6 both mediate extensive mRNA poly(A) tail trimming and degradation**. Rrp6, in contrast to Dis3, confers only one exosome nucleolytic activity that is not essential. To determine the full extent of exosome activity, we depleted Dis3 or Dis3 and Rrp6 simultaneously from the nucleus for 40 min at 30 °C using the AA system[39], which, as expected, stabilized the *NEL025c* CUT (Supplemental Fig. 4a, green series). Accordingly, our DRS dataset revealed higher levels of CUTs and snRNA and rRNA precursors in *DIS3-AA* and *DIS3-/RRP6-AA* cells. In the double-depleted strain, CUTs had poly(A) tails of 56 As and snRNAs and rRNAs tails of 63 As (Fig. 4a). At some snRNA loci, transcription termination can be mediated by both CPF and NNS[44]. Thus, these long poly(A) tails may be attributable to Pap1 activity. However, we found that poly(A) tails of nuclear ncRNAs were ~37 As in the *pap1-1 rrp6Δ* strain (Fig. 4a), confirming that TRAMPs can produce poly(A) tails that are comparable in lengths to mRNAs.

The mean mRNA poly(A) tail length increased from 40 As in WT to 47 and 56 As in *DIS3-AA* and *DIS3-/RRP6-AA* cells, respectively (Fig. 4b, c). The vast majority of moderately to minimally expressed mRNAs were affected, regardless of their poly(A) tail length in the WT background. In contrast, poly(A) tail length of highly expressed mRNAs was less affected (Fig. 4b, d and Supplemental Fig. 4b–e). Accordingly, the poly(A) tail length of *TDH3* mRNA was mostly unchanged in *DIS3-AA* and *DIS3-/RRP6-AA* cells, similarly to the *rrp6Δ* background. While all depletions markedly increased the poly(A) tail length of *RPL18B* mRNA (Fig. 4e and other examples Supplemental Fig. 4f), that change occurred without major alterations of the efficiency of poly(A)+ RNA extraction (Supplemental Fig. 4a, right). A large proportion of mRNAs was upregulated in Dis3- and Dis3/Rrp6-depleted cells, but again transcript stabilization did not correlate with poly(A) tail length change (Fig. 4d). Since the loss of a cytoplasmic exosome cofactor, Ski2 helicase, did not lead to substantial mean poly(A) tail changes but only to minor

alterations in mRNA stability (Fig. 4b, d and Supplemental Fig. 4g, h), we conclude that Dis3 and Rrp6 regulate mRNA stability and poly(A) tail length, mainly in the nucleus.

**PAN2/3 and CCR4-NOT cytoplasmic deadenylases target both distinct and overlapping sets of transcripts**. Cytoplasmic mRNA decay is regulated by CCR4-NOT and PAN2/3 deadenylase complexes. We sequenced poly(A)+ RNA from strains deleted for the catalytic subunits of each complex, *ccr4Δ* and *pan2Δ*. A substantial elongation of mRNA poly(A) tails was observed in *pan2Δ* and *ccr4Δ* backgrounds by an average of 7.17 and 6.66 As, respectively, with no major change in expression levels (Fig. 5a and Supplemental Fig. 5a, b). Surprisingly, mean poly(A) tail changes in both strains were poorly correlated, indicating that PAN2/3 and CCR4-NOT activities are directed towards different sets of substrates (Fig. 5b). Accordingly, Pan2 appeared to target mRNAs of high and medium abundance preferentially, and its impact on poly(A) tail length decreased together with transcript expression levels. The opposite trend was observed for Ccr4 (Fig. 5c and Supplemental Fig. 5a, b). Given that highly expressed mRNAs consisted of ~60% of all mRNA molecules in the library (Supplemental Fig. 5c), at steady-state PAN2/3 appears to be the main cellular deadenylase with the lowest transcript specificity, whereas Ccr4 is more target-specific. To further validate our general conclusions, we inspected polyadenylation profiles of selected mRNAs. As previously shown by LM-PAT, the poly(A) tail lengths of three well-expressed mRNAs, involved in septin organization (*CDC42*, *APQ12*, and *SHS1*), depended on CCR4-NOT and/or PAN2/3[45] (Supplemental Fig. 5d). In our DRS dataset, the highly expressed *TDH3*, *RPS28B*, and *RPP1B* mRNAs were exclusively targets of Pan2 (Supplemental Fig. 5e), while *HTA1*[46], *ALD5*, and *UTP22* mRNAs were Ccr4-specific (Supplemental Fig. 5f and for other examples, see Supplemental Fig. 5g).

Finally, we inspected the nuclease sensitivity of all mRNA poly(A) tails and found that 61.8% of all annotated mRNAs and 82.6% of mRNAs that could be detected by DRS were substrates to either nuclear and/or cytoplasmic deadenylation (Fig. 5d). These results collectively indicate that the regulation of mRNA poly(A) tail length by exonucleolysis is a multistep, substrate-specific process that occurs in the nuclear as well as cytoplasmic compartments.

**Transcript poly(A) tail length strongly depends on growth conditions**. Growth conditions alter both the rates of mRNA transcription and decay[47]. We compared poly(A) tail length profiles of cells grown in rich media at a steady state at 25 and 30 °C with cells subjected to heat stress at 38 °C for 12 min or 37 °C for 1 h (Fig. 6a and Supplemental Fig. 6a–c). At 30 °C and 25 °C, mRNA abundance and poly(A) tail length were comparable. In contrast, 12 min of heat stress decreased the abundance and shortened poly(A) tails of highly expressed mRNAs. Curiously, 1 h of incubation at 37 °C increased the mRNA abundance of highly and moderately expressed mRNAs and globally elongated poly(A) tails compared with 25–30 °C conditions. To examine some functional groups, we selected mRNAs that were either up- or downregulated by heat stress more than fivefold (Supplementary Data 1, Fig. 6b and Supplemental Fig. 6a–d). We observed that upregulation was accompanied by poly(A) tails elongation, whereas downregulated transcripts had, on average, shorter tails. At a steady state of 30 °C, similar poly(A) length shifts were observed, although expression levels were unchanged when compared with 25 °C. Finally, downregulated mRNAs seem to strongly rely on exonucleolysis for their regulation of abundance since 88% of them were targets of either the exosome or

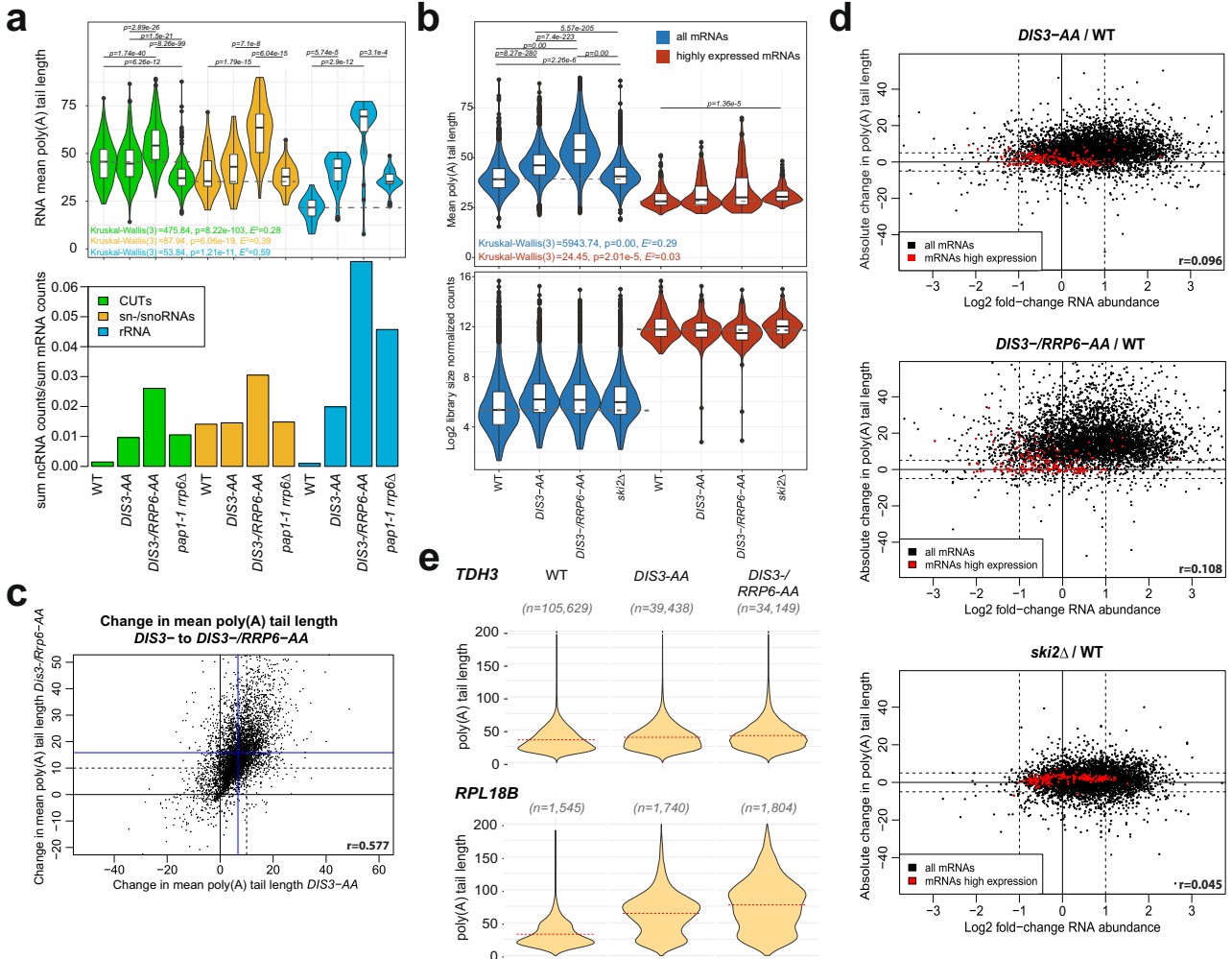

**Fig. 4 Dis3 and Rrp6 mediate extensive mRNA poly(A) tail trimming. a** Violin plot of mean poly(A) tail lengths of CUTs, snRNAs, and rRNAs detected in WT, *DIS3-AA*, *DIS3-/RRP6-AA*, and *pap1-1 rrp6Δ* cells (top), and the sum of all DRS counts that map to those ncRNA loci, normalized to the sum of counts mapped to protein-coding loci (bottom). The number of RNAs detected n = [WT, *Dis3-AA*, *Dis3-/Rrp6-AA*, *pap1-1 rrp6Δ*] was for CUTs = [143, 436, 579, 434], sn/snoRNAs = [36, 52, 59, 59], and rRNAs = [17, 23, 24, 23]. For the upper panel, Kruskal–Wallis test *P* value and Holm-corrected *P* values for two-sided Dunn pairwise test are shown. Dashed gray lines show the mean values for WT strain to ease visual comparison. Box plots show median value (solid bold line) and 1st and 3rd quartiles, whiskers represent 1.5 × IQR (interquartile range). Outliers are marked as black points. **b** Mean mRNA poly(A) tail lengths (top panel), and log2 mRNA abundance (bottom) for WT (n = 5079), *DIS3-AA* (n = 5035), *DIS3/RRP6-AA* (n = 5109) and *ski2Δ* (n = 5081) strains. Data for highly expressed mRNAs are shown in red. For the upper panel, Kruskal–Wallis test *P* value and Holm-corrected *P* values for two-sided Dunn pairwise test are shown. Dashed gray lines show the mean values for WT strain to ease visual comparison. Box plots show median value (solid bold line) and 1st and 3rd quartiles, whiskers represent 1.5 × IQR (interquartile range). Outliers are marked as black points. **c** Relationship between absolute change in mean poly(A) tail length between *DIS3-* and *DIS3-/RRP6-AA* cells compared with WT. Blue lines designate the mean poly(A) tail length change in each strain. Dashed lines mark a 10-adenosine change of mean poly(A) tail length. **d** Relationship between absolute change in mean mRNA poly(A) tail length and log2 fold-change mRNA abundance for *DIS3-AA*, *DIS3-/RRP6-AA*, and *ski2Δ* cells, respectively, compared with WT. Dashed lines show a twofold change in expression and a 5-adenosines change of poly(A) tail length. **e** Violin plots of the poly(A) tail length distribution of DRS reads for *TDH3* and *RPL18B* mRNAs for WT, *DIS3-AA*, and *DIS3-/RRP6-AA*. The number of DRS reads for each condition is shown for each transcript. Red dashed lines show the mean for each condition. Pearson's correlation coefficients are indicated on each scatterplot.

deadenylases, compared to 34% in the case of upregulated ones (Fig. 6c).

To evaluate the impact of nutrient content on mRNA and poly(A) tail metabolism, we compared DRS datasets from cells grown in a rich medium containing glucose as the carbon source with those from a minimal medium that contained raffinose and galactose (Fig. 6d). At 25 °C in minimal medium, mRNAs, regardless of abundance, generally harbored short poly(A) tails, so the anti-correlation between the lengths of poly(A) tails and expression levels evident in rich media was virtually lost (Fig. 6d, compare black and blue series and Supplemental Fig. 6e). This

effect was not attributable to a decrease in Pap1 activity because poly(A) tails of newly produced mRNAs had a comparable size, regardless of growth conditions (Fig. 2a, b and Supplemental Fig. 2a). Highly expressed mRNAs constituted a smaller fraction of all mRNA counts in cells grown in the minimal medium compared to the rich medium (Supplemental Fig. 6e). The abundance of these mRNAs further decreased after heat shock in a minimal medium, albeit without a strong impact on poly(A) tail length. Collectively, growth conditions, especially the nutrient source, had a major impact on mRNA expression and poly(A) tail length profiles.

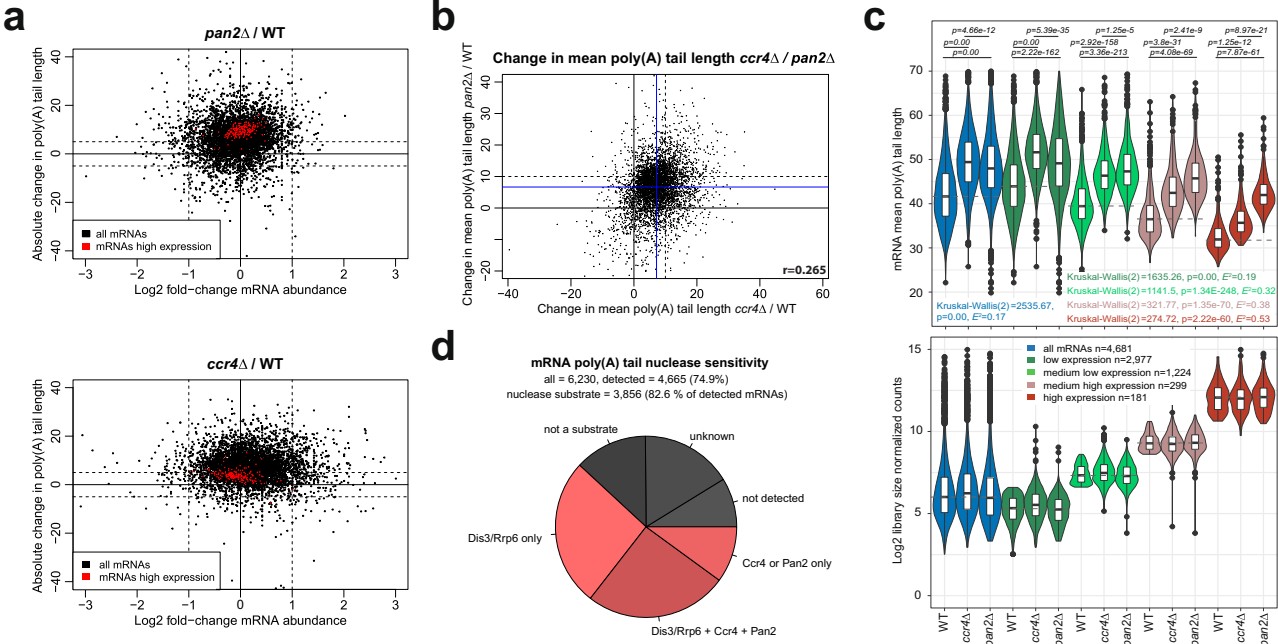

**Fig. 5 PAN2/3 and CCR4-NOT cytoplasmic deadenylase complexes can target distinct sets of transcripts. a** Relationship between log2 fold change in mRNA abundance and absolute change in mean poly(A) tail length for *pan2Δ* (top) and *ccr4Δ* (bottom) strains relative to WT. Dashed lines show a twofold change in expression and a 5-adenosine change of mean poly(A) tail length. **b** Relationship between absolute change in poly(A) tail length in *ccr4Δ* cells relative to WT compared with an absolute change in poly(A) tail length relative to WT for *pan2Δ*. Blue lines designate the mean poly(A) tail length change in each strain. Dashed lines mark a 10-adenosine change of mean poly(A) tail length. **c** The violin plot in the top panel shows the distribution of mean poly(A) tail lengths of detected mRNAs, binned by expression level in WT. The bottom panel shows log2-scaled mRNA abundance. The sum of counts from each mRNA category is indicated. Statistical significance for each expression group was calculated using the Kruskal–Wallis test *P* value and Holm-corrected *P* values for the two-sided Dunn pairwise test. Box plots show median value (solid bold line) and 1st and 3rd quartiles, whiskers represent 1.5 × IQR (interquartile range). Outliers are marked as black points. **d** Pie chart showing in red the fraction of cellular mRNAs that were deadenylated by only the exosome (1643), only CCR4-NOT and/or PAN2/3 (625), or all three complexes (1588). Out of a total of 6230 annotated mRNAs, 4665 were reproducibly detected. Transcripts that were unaffected or could not be detected by DRS are highlighted in gray.

## Discussion

Our comprehensive DRS analysis unveiled mechanisms of post-transcriptional mRNA and ncRNA poly(A) tail biogenesis, which was regulated in response to growth conditions, and demonstrated that nuclear and cytoplasmic nucleases play an important role in this process (Fig. 7).

The biogenesis of ncRNA poly(A) tails remained unclear because both polyadenylation and degradation act in parallel. The widely accepted model postulated that TRAMP4/5-mediated oligoadenylation enhances decay by recruiting the exosome/Rrp6 complex, whereas the addition of a long poly(A) tail by Pap1 promotes stability and export. This model does not fully align with our results, which showed that nuclear ncRNAs have poly(A) tails lengths comparable to mRNAs, even in Pap1-impaired cells. The TRAMP4/5, exosome, and NNS complexes interact both functionally and physically[1,48] but do not form a large stable assembly. The action of each of these complexes should instead be considered in a probabilistic manner. TRAMP4/5 recruitment can thus increase the likelihood of decay but does not assure it. Moreover, although TRAMP is not a processive enzyme[7,8], it can still be recruited for a second or third round of adenylation or, as shown by our results, target Pap1-adenylated substrates. Furthermore, in vitro studies showed that NNS RNA-binding activity stimulates TRAMP4 adenylation by stabilizing the complex on the substrate[1], supporting the synthesis of long poly(A) tails, at least on NNS targets. The addition of a long poly(A) tail can be required to initiate decay on highly structured substrates, such as rRNA precursors because the core exosome channel leading to the Dis3 active site accommodates a 25–30 nt long RNA[49]. Short,

3–5-adenosine long, poly(A) tails often detected by RNA-seq at ncRNAs[3,9] rather represent a trace of ongoing decay. However, each full-length ncRNA can undergo a phase of multiple TRAMP-adenylation cycles before decay is initiated. The relative amount of these highly polyadenylated species is a function of the time between transcription termination and decay initiation.

The nuclear biogenesis of mRNA poly(A) tails is seemingly noncontroversial. Cleavage and polyadenylation are coupled and thought to cooperate tightly with nuclear export[17]. Thus, mRNAs appear to escape nuclear degradation and be subject only to cytoplasmic decay. This view stems from the provocative gene-gating hypothesis that was proposed by Blobel in 1985[50] and proven true only for some mRNAs upregulated in response to growth conditions[51]. Single-molecule transport studies showed that, regardless of gene localization, mRNAs can diffuse freely within the interchromatin space for long periods before export[17,52]. Depending on the nuclear localization of genes, some mRNAs can thus have longer nuclear dwell periods that allow ample time to become targets of the exosome and its TRAMP4 and TRAMP5 cofactors. We found that mRNAs with low and medium expression were the most prominent exosome/TRAMP4/5 targets. We hypothesize that the export of those mRNAs is less efficient than highly expressed transcripts. Indeed, gene-gated *HSP104* expression is only regulated by TRAMP4/exosome-mediated decay in export-impaired *mft1Δ* cells[40]. We also speculate that exosome-mediated exonucleolysis and TRAMP4/5 re-adenylation could influence the export rate of some mRNAs, for which a strict length of the poly(A) tail is required[14] (Fig. 7).

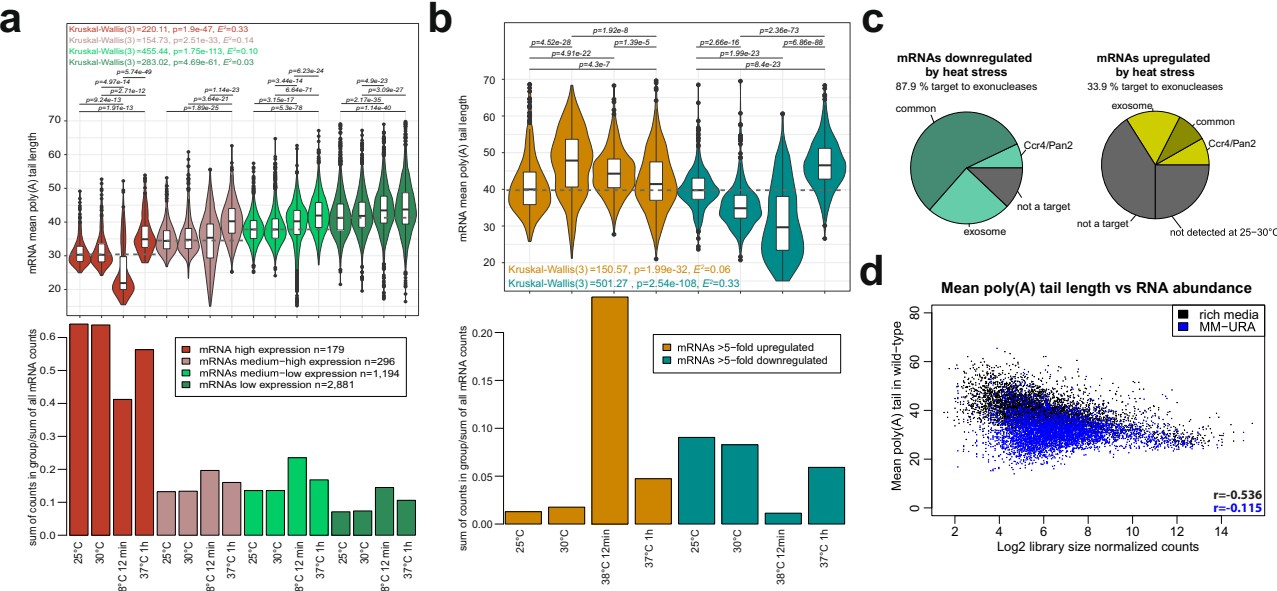

**Fig. 6 Transcript poly(A) tail length depends on growth conditions. a** Violin plots in the top panel show the mean poly(A) tail length distribution across mRNAs from wild-type cells that were grown at different temperatures, binned by expression level in WT. The bottom panel shows the fraction of mRNAs DRS counts for each expression bin in a given condition. Box plots show median value (solid bold line) and 1st and 3rd quartiles, whiskers represent 1.5 × IQR (interquartile range). Outliers are marked as black points. **b** Same as (**a**) but for mRNAs that were at least fivefold upregulated (orange series) or downregulated (green series) following heat shock. Box plots show median value (solid bold line) and 1st and 3rd quartiles, whiskers represent 1.5 × IQR (interquartile range). Outliers are marked as black points. **c** Pie charts showing the nuclease sensitivity of heat-regulated mRNAs, which are listed in Supplementary Data 1. **d** Relationship between mRNA abundance and mean poly(A) tail length for cells that were grown in rich media (black, same as Fig. 1a) and MM−URA (blue) at 25 °C as explained in the main text.

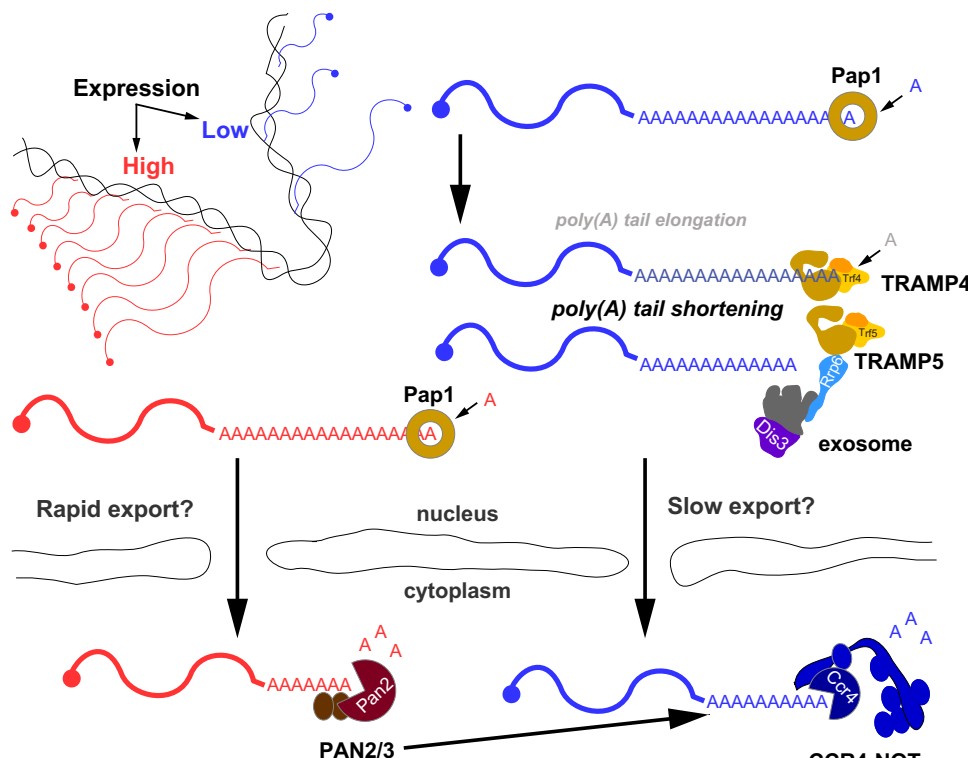

**Fig. 7 mRNA poly(A) tail length is extensively regulated post-transcriptionally in both the nucleus and the cytoplasm.** For details, see "Discussion".

Our data revealed that at steady state, both the CCR4-NOT and PAN2/3 cytoplasmic deadenylase complexes have a large non-overlapping set of targets and act redundantly on shared substrates representing only 10.9% of detected mRNAs. This refines the biphasic mRNA deadenylation model that postulates that initial deadenylation is mediated by PAN2/3 followed by CCR4-NOT[20,24]. Our findings favor the notion that in the initial phase, both complexes act independently according to their substrate specificity and that initial deadenylation is unlikely to be coordinated between these two complexes. PAN2/3 substrate recruitment is nonspecific because it relies on polyadenosine and Pab1 binding[20,21,53]. Highly expressed mRNAs constitute ~60% of all protein-coding molecules, and these are the most common PAN2/3 targets. In contrast, the CCR4-NOT complex, in addition to Pab1 binding[22], has been shown in different organisms to be targeted to its substrates via adaptor proteins that can also accelerate deadenylation[54–56], thus making this deadenylase complex more likely targeted to mRNAs of low expression as evidenced by our datasets. CCR4-NOT also directly stimulates decapping[57], suggesting that recruitment of this complex is more tightly linked to subsequent decay. However, this aspect cannot be fully addressed with our data due to constraints in detecting non- or oligoadenylated RNAs.

## Methods

**Yeast strain culture.** The yeast strains that were used in this study are listed in Supplemental Table 1. The yeast strains were grown in YPDA media, with the exception of *MEX67-AA* strains that contained URA-plasmids, which were grown in minimal media that lacked uracil with 2% raffinose and 2% galactose. All of the cultures were harvested in the logarithmic growth phase. To perform heat shock, the W303 WT, *mft1Δ*, and *MEX67-AA* strains were pre-grown at 25 °C. The cultures were complemented with an equal volume of media that was preheated to 51 °C and incubated at 38 °C for 12 or 15 min for *mft1Δ* and *MEX67-AA* series, respectively. Afterward, the cells were inactivated by adding an equal volume of 96% ethanol that was cooled to −20 °C. Precultures for the W303 *pap1-1* series were prepared at 25 °C. The cells were then diluted and shifted to 30 °C for 20 h. The deadenylase mutant strains *ccr4Δ*, *pop2Δ*, and *pan2Δ* mutant together with a WT strain, were grown continuously at 30 °C. The *rrp6Δ*, *trf4Δ*, *trf5Δ*, and *rrp6Δ trf4Δ* exosome mutants together with WT strains were grown at 30 °C. The *DIS3-AA*, *DIS3-/RRP6-AA*, and *ski2Δ* were grown at 30 °C and the anchor-away strains were supplemented with rapamycin (Cayman Chemicals cat. no. 13346) to a final concentration of 1 μg/ml for 40 min before harvest. A WT W303 strain was pre-grown at 25 °C and then transferred for 1 h of growth in a water bath with shaking at 37 °C.

**RNA preparation, reverse transcription, and qPCR analysis.** RNA from yeast cells was extracted using the hot acid phenol method. Cell pellets from 10 to 30 ml cultures at OD 0.3 to 0.6 were resuspended in 400–600 μl of TES buffer (10 mM Tris [pH 7.5], 5 mM ethylenediaminetetraacetic acid, and 1% sodium dodecyl sulfate [SDS]) and complemented with an equal volume of acid phenol (catalog no. P4682, Sigma-Aldrich). The samples were incubated at 65 °C with vortexing in two rounds for 40 and 25 min. Between and after incubations, the samples were centrifuged at $20,000 \times g$ for 10 min at 4 °C. The resulting supernatant was vortexed with an equal volume of chloroform (catalog no. 112344306, Chempur) and centrifuged again. The RNAs from the supernatant were precipitated using 96% ethanol and a 65 mM final concentration of LiCl.

Yeast polyadenylated RNA for DRS library construction and reverse-transcription qPCR analysis was extracted from 35 to 75 μg total RNA using the ThermoFisher dynabeads oligo-(dT)$_{25}$ kit (catalog no. 61002 or 61005 according to the manufacturer's protocol. In brief, 35 μg of RNA was resuspended in 50 μl of water and mixed with 50 μl of Binding Buffer (20 mM Tris-HCl, pH 7.5, 1.0 M LiCl, 2 mM EDTA). This sample was denatured for 2 min at 65 °C and then put on ice for 2 min. Then the sample was mixed with 50 μl of beads pre-washed in Binding Buffer (slurry beads volume per one sample was 100 μl). The samples were incubated 15 min at room temperature and then washed two times with 100 μl of Wash Buffer (10 mM Tris-HCl, pH 7.5, 0.15 M LiCl, 1 mM EDTA 10 mM Tris-HCl, pH 7.5). Buffer was removed from dynabeads, which were then resuspended in 10-12 μl of water and eluted 2 min at 80 °C.

The total RNA fraction was treated with DNAse (Turbo-DNase, Ambion) according to the manufacturer's protocol. Total and poly(A)$^+$ fractions were used for reverse transcription with (dT)$_{18}$ and random hexamers using SuperScript III (Invitrogen, cat. no. 18080093) as specified in the manufacturer's protocol. cDNA was diluted 10–20 times and used for qPCR analysis with the Platinum SYBR Green qPCR Super-Mix-UDG kit (Invitrogen, cat. no. 11733-046) and LightCycler 480 (Roche). The list of primers that were used for qPCR analysis is specified in

Supplemental Table 2. The relative cDNA concentration was calculated with the 2nd derivative maximum method and normalized for quantities and dilutions that were used in each experiment separately. Each reaction was carried with two or three biological replicates, each measured in duplicate. Analysis of qPCR results is deposited at Mendeley data (https://doi.org/10.17632/v5vm3dmm8y.1).

**Preparation of standards with predefined poly(A) lengths.** Templates for spike-ins were prepared in two consecutive PCR reactions. First, an 827-bp fragment of Renilla luciferase from pRL5Box plasmid was amplified using RLucF1 and RLucR1 primers. The purified amplicon was used as a template in the second PCR reaction with RLuc_T7_F2 primer containing the T7 promoter sequence, and backward primer RLuc_Ax_R2 introducing poly(A) tail of a defined length (from 10 to 90 As). The resulting PCR product was analyzed and purified by gel electrophoresis. In vitro transcription reaction was performed at 37 °C for 1.5 h in a 50 μl reaction volume containing: 600 pmols T7 template, 10 μl of 5× transcription buffer (200 mM Tris-HCl, 30 mM MgCl$_2$, 10 mM spermidine, 50 mM NaCl), 5 μl of rNTPs mix (20 mM each), 5 μl of 100 mM DTT, 0.5 μl of 1% Triton X-100, 80U ribo-nuclease inhibitor, 100U T7 RNA polymerase. Then, the DNA template was digested by adding 4U of TURBO DNase (Ambion) for the next 15 min. The reaction was stopped with 2.5 μl of 0.5 M EDTA pH 8.0 and following phenol/chloroform extraction and ethanol precipitation. The quality of spike-ins RNA was visually assessed by denaturing electrophoresis. Spike-ins were purified on RNA purification beads (Kapa Pure beads) and subjected to nanopore sequencing.

**Northern blot analysis.** To analyze *snR13* snoRNA, 5 μg of the total RNA or 100 ng poly(A)$^+$ RNA was separated by electrophoresis in 6% denaturing polyacrylamide–8 M urea gel in 1× TBE, followed by RNA immobilization on a Hybond N + nylon membrane (Amersham, cat. no. RPN82B) by wet electro-transfer in 0.5× TBE. RNA was fixed by ultraviolet cross-linking. Hybridization was performed in PerfectHyb Plus hybridization buffer (Sigma-Aldrich, cat. no. H7033). The blot was handled according to standard procedures and probed with a DNA oligonucleotide that was labeled at the 5′-end with T4 PNK (New England Biolabs, cat. no. M0201L) and [γ-$^{32}$P]adenosine triphosphate (Hartmann Analytic) at 42 °C overnight. After hybridization, the membrane was washed twice for 30 min with 2× SSC and 0.1% SDS at 42 °C and then exposed to a PhosphorImager screen (FujiFilm), which was scanned using a FLA 7000 scanner (FujiFilm).

To analyze *TDH3* and *RPL18B* mRNAs, 20 μg of the total RNA or 100 ng poly(A)$^+$ RNA was pre-annealed to 100 ng DNA oligonucleotide, directed to the respective transcript 3′-UTR and subsequently digested with 5 U RNaseH (New England Biolabs, cat. no. M0297S) in a buffer that contained 20 mM Tris-HCl (pH 7.5), 100 mM KCl, 10 mM MgCl$_2$, 5% sucrose (w/v), and 10 mM DTT. Following digestion, RNA was extracted with phenol:chloroform and chloroform alone and precipitated with 99% ethanol in the presence of 0.3 M sodium acetate (pH 5.2) and GlycoBlue (ThermoFisher Scientific, cat. no. AM9515) coprecipitate. Pellets were resuspended in deionized H$_2$O, and the further steps were analogous to *snR13* snoRNA. Between hybridizations, the probes were stripped off the membranes at 65 °C by boiling with 0.1% SDS.

Northern blot signals were quantified using ImageJ from two biological repeats. For better comparison with DRS datasets, the overall signal was normalized to the intensity of the region corresponding to the main band on the northern blot.

**Nanopore sequencing.** Direct RNA sequencing was performed as described by Bilska et al.[58]. Briefly, RNA libraries were prepared from 500 ng human, murine, *A. thaliana*, or *C. elegans* cap-enriched mRNA mixed with 50–200 ng oligo-(dT)$_{25}$-enriched mRNA from *Saccharomyces cerevisiae* yeast with a Direct RNA Sequencing Kit (catalog no. SQK-RNA002, Oxford Nanopore Technologies) according to the manufacturer's instructions. Sequencing was performed using R9.4 flow cells on a MinION device (ONT). Raw data were basecalled using Guppy (ONT). Raw sequencing data (fast5 files) were deposited at the European Nucleotide Archive (ENA, (for a list of accession numbers, see Supplemental Table 3). The full list of sequencing runs, with appropriate summaries, is shown in Supplementary Data 2.

## Bioinformatic analysis

*Preparation of yeast genome annotation.* Custom annotation of the yeast genome was used to map the sequencing reads (file name: SacCer3_custom_annotation_ORFs_ncRNA_CUTs_SUTs_XUTs.bed). The annotation was prepared by merging published datasets from David et al.[59]; file name: Steinmetz_annotations_sacCer3_latin_sorted.bed), van Dijk et al.[60]; file name: van_Dijk_2011_-XUTs_V64.bed), and sgd_other from the University of California Santa Cruz (UCSC) Genome Browser. ORF-T features that were not listed in David et al.[59] but are specified as ORF-T features in the UCSC Genome Browser (Table Browser and track sgdGenes) were added to the custom annotation. Importantly, these missing ORF-T features were not specified as noncoding in the David et al.[59] annotation. XUT features from van Dijk et al.[60] and CUT and SUT information from David et al.[59] were merged into the custom file, and overlapping feature IDs were merged using the bedtools merge function. The sgd_other track from the UCSC Genome Browser was used to find other noncoding features. The noncoding features that were shorter than 50 bp were filtered away because the technology does not allow the mapping of reads of this length. LTR, transposon, tRNAs, and replication

origins were also discarded from the annotation. This procedure extracted 93 additional features. The resulting custom annotation contained 9105 features, including 2870 noncoding ones and is available at Mendeley data (https://doi.org/10.17632/v5vm3dmm8y.1).

*Publicly available datasets.* The dataset containing calculated quantitative measures of translation for 4621 yeast genes was obtained from Supplemental Table S1 from Siwiak and Zielenkiewicz, 2010[38]. Expression values of transcripts from poly(A)+ or total RNA fraction, described in Presnyak et al.[29], were downloaded from GEO (GSE57385, files: GSE57385_PolyA_FPKM.txt.gz and GSE57385_RiboZero_FPKM.txt.gz). Tiling array data[28] were obtained from Domenico Libri and are publicly available at (E-MTAB-1246). Protein abundance estimates per cell were obtained from Supplementary Information Table 4 from Ho et al.[37]. Poly(A) length measurements with the PALseq method[30] were downloaded from GEO (GSE52809, file: GSE52809_Cerevisiae_total.txt.gz). RATE-seq data were obtained from Supplemental Table S5 from Neyomotin et al.[34], with the kind assistance from M. Schmid. Data regarding RNA PolII density were obtained from Supplemental Table S1 from Pelechano et al.[35] (column: RNA pol II density (per kb)), with the kind assistance from M. Schmid. cDTA data from Sun et al.[36] were obtained from ArrayExpress: E-MTAB-760, with the kind assistance from M. Schmid. Short-read (dT)-primed RNA-sequencing data were obtained from data deposited by Schmid et al.[31] and Tudek et al.[32] with accession numbers GEO: GSE108477 and GEO: GSE108550. Samples used were the average of one replicate of Mex67-AA and Nab2 input RNA -rapamycin with or without in vitro polyadenylation and ribodepletion. Only signal around the cleavage site was used and normalized to *S. pombe* spike-in as described in the original publications.

*Poly(A) lengths determination.* Yeast-originating DRS reads were separated from the other samples by mapping to the respective reference transcriptomes using Minimap2 2.17[61] and filtering out reads that were unmapped to the yeast reference. The obtained reads were mapped to reference transcript sequences (described above) using Minimap 2.17[61] with options -k 14 -ax map-ont –secondary = no and processed with samtools 1.9 to filter out supplementary alignments and reads mapping to reverse strand (samtools view -b -F 2320). The poly(A) tail lengths for each read were estimated using the Nanopolish 0.13.2 polya function[25]. In subsequent analyses, only length estimates with the QC tag that was reported by Nanopolish as PASS were considered. Since the replicates strongly correlated with one another repeats were summed. Tables with the number of counts, mean, median, and geometric mean poly(A) tail lengths are deposited at Mendeley data (https://doi.org/10.17632/v5vm3dmm8y.1).

The same procedure for mapping and poly(A) length estimation was applied to predefined poly(A) standards, but reads were mapped to Renilla luciferase sequence instead.

*Data analysis and visualization.* Analyses of changes in mean poly(A) tail length and RNA abundance were performed using R and visualized using the ggplot2 package[62,63]. Box plots (within violin plots representing RNA poly(A) lengths distribution or expression level) show median value (solid bold line) and 1st and 3rd quartiles, whiskers represent $1.5 \times$ IQR (interquartile range). Outliers (data points that do not fall within $1.5 \times$ IQR) are marked as black points. Transcripts that were detected by ≤5 reads were removed from the datasets by setting their value to 0.01 for count number and poly(A) tail length alike, to allow their inclusion in the comparisons. Counts for each transcript were normalized to library size or in the case of the *pap1-1* series library size and qPCR-derived coefficient. For the purpose of graphical display, library size normalized counts were multiplied by 1,600,000, which is the approximate maximum number of *S. cerevisiae* mapped reads that were obtained from the largest DRS dataset.

*Binning mRNAs by the expression level.* The binning of mRNA by expression level was performed on mRNAs detected in each set of strains by setting the following cutoffs: (1) highly expressed mRNAs: log2(library normalized counts) ≥ −10, (2) moderately to highly expressed mRNAs: −12 < log2(library normalized counts) ≤ −10, (3) moderately to minimally expressed mRNAs: −14 < log2(library normalized counts) ≤ −12, (4) minimally expressed mRNAs: log2(library normalized counts) ≤ −14.

To produce violin plots for mean poly(A) tail lengths and log2 mRNA abundance of mRNAs and ncRNAs, only features that were detected in each dataset were used. The barplots showing the sum of counts for ncRNAs include all features that were annotated in each group. This approach accurately showed poly(A) tail lengths, while informing about a possible loss of a fraction of ncRNAs in terms of abundance.

*Identification of oligoadenylated mRNA fraction.* To calculate the number of mRNAs with a large oligoadenylated fraction, nine replicates of poly(A)+ fraction DRS from WT strain were averaged (samples: WT BY4741 30 °C—three replicates, WT W303 30 °C control to *pap1-1*, DIS3-AA series and *ski2Δ*—two replicates, WT W303 30 °C control to deadenylase series—two replicates, WT W303 25 °C—two replicates) and used as a reference for DRS prepared on the total RNA fraction, which was a mean of two biological and one technical replicate. In all

analyses, only transcripts detected in all datasets were considered. Based on the cross-comparison of the percentage recovery of selected mRNAs by qPCR and DRS total versus poly(A)+ fraction DRS analyses we first selected mRNAs that were more than 1.2-fold upregulated in the total compared to the poly(A)+ fraction. Upregulation is to be expected if additional reads are captured in the total compared to the poly(A)+ fraction. As additional oligo-tailed reads should decrease the mean poly(A) tail estimate we selected from upregulated mRNAs those that displayed a decrease in mean poly(A) tail estimate. The precision of mean poly(A) tail estimation decreases with the expression level, therefore we have calculated separate cutoffs for four expression categories (defined in the section above). Those values are the standard deviations of mean poly(A) tail length estimation in each mRNA abundance group. (1) highly expressed mRNAs: cutoff −1.727579 adenosines, (2) moderately to highly expressed mRNAs: cutoff −2.080906 adenosines, (3) moderately to minimally expressed mRNAs: cutoff −2.763902 adenosines, (4) minimally expressed mRNAs: cutoff −4.157491 adenosines. Collectively, the 1.2-fold-change cutoff combined with a change in mean poly(A) tail estimation in the total compared to poly(A)+ fits the data analyzed by qPCR in Supplemental Fig. 1j.

*Pearson's correlations.* Pearson's correlations and scatterplot matrices were produced with cor.test function in R from datasets that were filtered to include only values that were present in both tested conditions. For correlations with the mRNA synthesis rate, half-life, protein abundance, and protein synthesis rate, DRS datasets from two WT W303 cells that were grown at 25 and 30 °C and WT BY4741 cells that were grown at 30 °C in rich media were filtered to include only features that were detected in all three datasets and averaged.

*Selection of strong exosome and deadenylase targets.* To evaluate unique and common targets of the exosome and deadenylases, mRNAs that had mean poly(A) tail lengths that increased by at least ten adenosines compared to respective wild-type control were selected from the *DIS3-/RRP6-AA* dataset, and a combined list of shared and unique targets of *pan2Δ* and *ccr4Δ*. Please note that these criteria are much more stringent than the one applied to calculate the fraction of oligoadenylated transcripts. Analysis of the four WT strains available shows that nearly all poly(A) tail length estimates oscillate between + /− 10 adenosines (see the comparison of WT 25 °C with other WT replicates in Supplemental Fig. 6b). Thus a poly(A) tail elongation above this threshold is unlikely to be an artifact.

*Gene ontology analysis.* Gene ontology analysis included in Supplementary Data 1 was performed using a list of systematic names with http://geneontology.org using the *S. cerevisiae* reference for biological process.

*Statistics and reproducibility.* Statistical analysis was conducted on data from two or more biologically independent experimental replicates. Statistical analysis of quantitative data was performed using R unless otherwise stated. The statistical tests used in each instance are mentioned in the figure legends. P values are shown on respective plots only for statistically significant comparisons (P value < = 0.001). Data are presented as bar or violin plots, with mean values indicated, as indicated in the figure legends, and individual data points are shown when the number of observations is low (<10). Samples with clear technical failures during RNA isolation and sequencing were excluded from analyses.

Multi-group comparisons of poly(A) lengths were performed in the R environment with Kruskal–Wallis test (kruskal.test from R stats package), and subsequent pairwise comparisons with two-sided Dunn test (kwAllPairsDunnTest from PMCMRplus package). Obtained P values were corrected for multiple comparisons using the Holm method. Effect sizes for Kruskal–Wallis tests are shown as ranked epsilon squared (calculated using rank_epsilon_squared from R effect size package) and indicated on the plots ($E^2$ parameter).

**Reporting summary**. Further information on research design is available in the Nature Research Reporting Summary linked to this article.

## Data availability

The data supporting the findings of this study are available from the corresponding authors upon reasonable request. The nanopore sequencing data generated in this study have been deposited in the European Nucleotide Archive database under accession codes: ERS4936519, ERS3526067, ERS4936515, ERS5470421, ERS5470422, ERS4936517, ERS5465275, ERS4936520, ERS5464585, ERS5465276, ERS4936521, ERS5464586, ERS5458723, ERS5458724, ERS5458726, ERS5458727, ERS5458725, ERS5458728, ERS4936516, ERS4936518, ERS4936513, ERS4936512, ERS5464062, ERS5464063, ERS5464064, ERS5464065, ERS5458721, ERS5458722, ERS5464066, ERS5464067, ERS5084525, ERS5084526, ERS5084529, ERS5084530, ERS5465444, ERS5465445, ERS5465446, ERS5470413, ERS5465447, ERS5470414, ERS5465448, ERS3526064, ERS5465450, ERS5465451, ERS5465452, ERS5470417, ERS5470416, ERS5465453, ERS5465454, ERS5465455, ERS5465456, ERS5465457, ERS5470037, ERS5470039, ERS6477293, ERS6477294, ERS6477295, ERS6477287, ERS6477288, ERS6477289, ERS6477290, ERS6477291, ERS6477292, provided also in Supplemental Table 3. Other raw data are accessible at https://doi.org/10.17632/v5vm3dmm8y.1.

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

## Acknowledgements

We thank Joanna Kufel for providing some of the yeast strains and comments on the manuscript, Dominico Libri for advice concerning Gudipati et al.[28] datasets, Paweł Grzechnik for critical reading and Manfred Schmid for technical advice. This work was mainly supported by the TEAM/2016-1/3 Foundation for Polish Science grant (to A.D.). A.D. is also a recipient of the ERA Chair's position, funded by the EU (agreement no. 810425). S.M.'s work was supported by a Polish National Science Centre grant (no. 2020/38/E/NZ2/00372 to S.M.). Work in the R. Tomecki group was supported by a Polish National Science Centre grant (no. 2017/26/E/NZ1/00724 to R.T.). Funding for this work in the T.H.J. laboratory was provided by the Independent Research Fund Denmark and a FEBS Long Term Fellowship together with an EMBO Long-Term Fellowship (ALTF 328-2019) to M.T.

## Author contributions

A.T. designed and performed the experiments, analyzed the data, and wrote the manuscript. P.K. processed the DRS data, performed essential bioinformatic and statistical analyses, and edited the manuscript. S.M. implemented DRS and prepared all sequencing libraries. R.T. performed the Northern blot analyses, edited the manuscript, and provided funding. M.T. prepared *MEX67-AA* samples and edited the manuscript. KMK prepared DRS spike-ins. T.H.J. edited the manuscript and provided funding. A.D. conceptualized and coordinated the project, provided funding, and edited the manuscript.

## Competing interests

The authors declare no competing interests.
