## [Peer Review File · Nature Communications]

REVIEWER COMMENTS

Reviewer #1 (Remarks to the Author):

In this nice manuscript, the authors take advantage of a state of art direct RNA sequencing technique they developed, using nanopore sequencing, to determine genome wide, and in an unbiased way, the poly(A)-tail length of individual transcripts under various genetic backgrounds and growing conditions.

Obviously, this is only the beginning of the use of this technique and this manuscript somehow sets up the foundations of many further analyses that will be using this approach.

Yet, in this paper, the authors manage to already address a number of questions, which remained mainly unsolved or only analyzed with reporter transcripts, with all the bias that it inevitably introduce.

In particular, they analyze the consequences of the absence of each most relevant exonucleases, deadenylases and poly(A) polymerases. This reveals in particular for the first time the involvement of the Trf5/TRAMP-exosome complex in regulating the poly(A)-tail length of mRNAs. It also settle the long-standing question of the specific role of the two deadenylase complexes, the CCR4-NOT complex and the PAN2/3 complex.

I have only one minor comment. The authors explain that the approach cannot measure the length of poly(A)-tails shorter than ~15 nucleotides, and they state that "The fraction of adenylated mRNA that was recovered did not depend on expression levels or DRS poly(A)-tail length estimation, indicating that factors other than poly(A)-tail length (e.g., secondary structures) also determine recovery efficiency."

I wonder if this might not rather reflect the fact that important fractions of many mRNAs might be present in an essentially deadenylated or oligo-adenylated form (see in particular Presnyak et al., Cell, 2015; <http://dx.doi.org/10.1016/j.cell.2015.02.029>). It might be important to address this question, either experimentally on a few individual transcripts or may be simply by looking if the efficiency of polyadenylated transcript recovery does not correlate with the fraction of polyA- vs polyA+ mRNAs, as determined by Presnyak et al.

Likewise, I don't understand the y-axis in Suppl. Fig. 3C, right panel, since it shows DRS measurements of poly(A)-tail lengths that begin at zero. There might be a scale adjustment to be made here?

Reviewer #2 (Remarks to the Author):

In this manuscript, Tudek and colleagues proposed to investigate the role of different decay factors in determining the poly(A) tail length in budding yeast. The authors used direct RNA sequencing, developed by Oxford Nanopore Technology, to assess the distribution of poly(A) tail lengths and transcript abundance. A set of bioinformatic tools were then used to analyze the generated data and interpret all the correlations. Overall, the manuscript contains a rich amount of data and the authors attempt to assess the correlation between poly(A) tail length and mRNA expression from multiple angles. Although the title of this study mentions "polyadenylation landscape", the manuscript focuses rather on the poly(A)-tails (particularly their length) than the process of polyadenylation itself, which is misleading.

My principal and most important criticism of the novelty claims in this manuscript, which center around the fact that authors applied direct RNA sequencing, are undermined by the fact that poly(A) tails shorter than 15 As cannot be detected by this technology. Given that poly(A) tails of yeast mRNA

are already quite short, and in absence of validation by other techniques, this rather limits the interpretation of poly(A) tail metabolism as presented in this study.

Another major technical concern is that the sequencing of poly(A) tails in the background of depleted deadenylase enzymes in a corresponding strain ($\Delta ccr4$, $\Delta pop2$ and $\Delta pan2$) is a poorly designed experiment. It has been shown previously that Ccr4 is the principal deadenylase in budding yeast and this is confirmed in this study. However, depleting Pop2 leads to the concomitant depletion of Ccr4 and, consequently, disrupting the deadenylase activity of the Ccr4-Not complex as a whole. The authors somehow failed to take this into account and the data is thus very difficult to interpret.

The majority of sections of this manuscript should be re-written and presented in a sequential and coherent manner. The reader might find some difficulties in keeping up with the narration of results and the underlying conclusions. The Introduction is an accumulation of literature on RNA polyadenylation and degradation factors in yeast without any clear axis of interest related to the aim of the study and in a confusing way. There is no linkage between paragraphs and there are abrupt transitions from one mechanism to another. One example from the Results is already the first paragraph (lines 119-134). The text is focused on a supplementary figure which is supposed to support the key result presented in Figure 1, but less description is applied to Figure 1 itself. There are other instances of confusing and meandering writing.

Specific concerns:

Lines 75-7: "During its cytoplasmic lifespan, the poly(A) tail is gradually shortened by two main deadenylases PAN2/3 and CCR4-NOT".

This statement is misleading in many ways, not least because the work by Narry Kim laboratory had provided compelling evidence that it is the CCR4-NOT that acts as the principal factor responsible for poly(A) tail control in the cytoplasm. There is no evidence that it does so in concert with PAN2/3, which this statement implies. The work cited based on *in vivo* reporters has largely been replaced by a more comprehensive view based on measurements of poly(A) tail lengths in cells of the transcriptome.

Lines 105-5: "We found that mRNA poly(A)-tail lengths are generally not correlated with transcript translation efficiency..."

This is also misleading because there are no measurements of translational efficiency described in this study, such as those that may be obtained by ribosome profiling combined with deep sequencing.

Lines 142-2: "Such heterogeneity of poly(A)-tail lengths increased gradually with decreasing expression".

This statement, in its most direct interpretation, is simply not supported by the data presented because the estimates of mRNA levels are used as a proxy for mRNA expression and translation. This weakens the analysis, as there is a substantial number of mRNAs that can exist in a translationally silenced state. The authors should at the very least acknowledge this fact before drawing conclusions.

Lines 144-5: "We instead suggest that individual RNAs undergo different poly(A)-tail length control."

There is indeed a heterogeneity in poly(A) tail lengths as determined by the experimental approach the authors have taken. This is neither a surprising observation nor a novel one. There is no evidence in this study for differences in regulation.

Lines 149-150: "...rather than poly(A)-tail length, mRNA abundance is a function of transcription and synthesis rates."

What the authors observe, in fact, that there is no correlation between poly(A) tail lengths and mRNA half-life. The suggestion about the determinant of mRNA abundance remains purely speculative.

Lines 152-5: there is a confusion of concepts between the "poly(A)-tail of short length (30-40 nucleotides)" and "reduction of poly(A)-tail length (deadenylation)".

To state "This indicates that a substantial reduction of poly(A)-tail length might not necessarily lead to rapid decay" might be an erroneous assumption because no data shown until this point is assessing decay rate as a function of the deadenylation process. The RNAs are still at the steady-state. Furthermore, in the same paper used as reference (Eisen et al, Mol Cell 2020), the rate (or "speed") of deadenylation is a function of the "degree" because the speed increases below a specific degree of deadenylation.

Lines 157-8: "The number of protein molecules per cell correlated strongly with mRNA abundance and anticorrelated with mean poly(A)-tail length²⁷ (Fig. 1d), which can be attributed to co-translational mRNA deadenylation."

The fact that the length of the poly(A)-tail is anticorrelated with translation efficiency has already been shown by others. The authors speculate that the same trend is observed in the budding yeast. I use the term "speculate" because there is no direct experimental evidence in this paper to make claims regarding any kind of correlation/anti-correlation of the number of protein molecules per length of the mRNA poly(A) tail. The cited papers do not provide clear distinct direct evidence for this either. The text nor the figure legend provides no description of how the authors arrived at this conclusion.

Lines 199-203: the difference between pap1-1 rrp6 Δ and the other two (pap1-1 & pap1-1 trf4 Δ) is clearly significant (almost 50% recovery). It is not at all clear how the authors assessed the significance between averages of 6.86 and 7.66. Drawing this conclusion "Trf4 and Trf5 proteins can compensate for its activity" is speculative.

Lines 208-211 present another confusion of concepts. Deficient adenylation is one process and deadenylation is another.

When describing a result, it would be helpful to provide a reference to the details in the figures: "highly expressed mRNAs in these mutant strains were virtually identical to the WT control. In contrast, mRNAs that had alterations of mean poly(A)-tail lengths exhibited alterations of poly(A)-tail patterns (Fig. 2h, Supplemental Fig. 2f)" it is not at all not clear what is what.

Lines 300-1: " A large proportion of mRNAs was up-regulated in Dis3- and Dis3/Rrp6-depleted cells" is quite misleading because log₂ mRNA abundance in the bottom panel of Fig 4b does not show a significant change in mRNA levels in depleted cells.

Line 385: "This extensive regulation suggests poly(A)-tail length has, next to the adjustment of mRNA levels, an important impact on shaping protein expression." Leaving aside the fact that proteins are not expressed but the genes are, it seems that with this statement the authors appear to contradict themselves on line 149: "Comparisons of our DRS results with existing datasets revealed that rather than poly(A)-tail length, mRNA abundance is a function of transcription and synthesis rates".

The general feeling one gets from reading this manuscript is that the principal novelty lies in the application of new and exciting methodology to sequence mRNAs. It is not without its limitations and, unfortunately, this manuscript does not provide any new mechanistic insights into mRNA regulation. Apart from significant issues with the design and interpretation of the experiments as well as a general lack of novelty, the manuscript is poorly worded in quite a few places and contains a large number of typos, grammar, and syntax errors. I would encourage the authors to reassess the suitability of this work, in its present form, for publication in any venue.

Reviewer #3 (Remarks to the Author):

In this work, "Comprehensive picture of the RNA polyadenylation landscape in the yeast *Saccharomyces cerevisiae*", the authors investigated the transcriptome-wide changes of poly(A) tail lengths and transcript abundance under various growth conditions and selected mutants of *S. cerevisiae*. The authors use Oxford Nanopore Direct-RNA sequencing (DRS) to investigate poly(A) length and transcript abundance independent of possible amplification artefacts. Through their data they demonstrate that poly(A)-tail length is not directly coupled to transcript abundance or translation regulation, and dissect the mechanisms of poly(A)-tail maturation in the nucleus and cytoplasm of *S. cerevisiae* through analysing relevant enzyme knockout mutants.

The approach is suitable to demonstrate the interplay of various poly(A) processing factors in biological systems, and to show correlative relationships to RNA abundance. The use of Oxford Nanopore Direct RNA sequencing enables deep insights into transcriptomics regulations. While poly(A) measurements with this technology may have somewhat low precision and suffer from low count numbers, they allow transcriptome-wide analysis. The outlined data analysis seems reproducible, and the data is made accessible. With the power of *S. cerevisiae* knockouts, the authors shed light on the complex interplay of poly(A) processing machinery and the outcome of transcript abundance. It is thus a valuable addition to the field of RNA biology.

The conclusions drawn appear to be robust and supported by the presented data. In general the paper is well-written and well structured. The experiments follow logically from each other, and together support the conclusions of the article.

A possible exception is the claim that the maximum poly(A) length is around 200. The data presented for this claim as a biological limit is somewhat weak and should be supported by more than single-gene violin plots ending at 200. We suggest the authors provide values for transcriptome-wide maximal measured poly(A) tails and investigate possible alternative explanations for these measurements. Do all transcripts harbor these lengths, is it specific transcripts? Do these measurements happen often enough to exclude technical artifacts? Are there alternative methods that could verify a maximum poly(A) length?

List of Suggestions:

Major:

Line 166: "with the limit of 200 adenosines"

And Line 179-182: "regardless of growth conditions, the polyadenylation limit was around 200 adenosines ...(Fig 2b, Supplemental Fig. 2a)."

Figure 2b shows only one gene and violin plots under various conditions. These plots have an upper limit of 200, but it is unclear whether this is the limit of the actual underlying data. Supplemental Figure 2a. Shows similar type of plots for three other chosen genes. From these graphs it is not clear that 200 is the maximally measured poly(A) length throughout the whole datasets. It is furthermore not possible to judge whether 200 is an arbitrarily chosen upper limit in plotting, a technical limit of the method (both bioinformatically and/or technically through sequencing) or really stemming from biological information. To support this claim, further data should be provided, or the claim has to be weakened to not be confused with a biological fact.

Minor:

Violin plots of poly(A) tail length of single transcripts should be supplemented with the number of measurements that lead to that plot. The shape of violin plots can be significantly influenced by the amount of datapoints contributing to it. The reader cannot judge whether the shape is based from sufficient measurements.

Figure 3c has unreadable small font in the legend

Figure 3d: The northern plot quantification graph is not normalised to overall intensity. To be able to compare to the density plot of Nanopore data, each line should sum up to 100%. Instead, at the moment the blue line is above the red line at all times suggesting a higher general intensity level and precluding direct comparison

Figure 6b: The Figure description uses the words "induced" and "repressed" even though only transcript levels in correlation to posttranscriptional regulation were measured. The words are misleading as they suggest active induction or repression of transcription. Instead, "down-regulation/reduced" or "upregulation/increased" should be used

Line 40: Cleavage and Polyadenylation Factor should be abbreviated to (CPF) not (CFP)

Line 43: "facilitates" should be "facilitate"

Line 385-387: "This extensive regulation suggests poly(A)-tail length has, next to the adjustment of mRNA levels, an important impact on shaping protein expression." The article does not provide any data supporting the claim that extensive poly(A) tail regulation has an impact on shaping protein expression. Actually, the initial claims of the paper rather support the notion that poly(A) length and translation are not coupled in a way that clearly predicts an outcome of one another.

Line 451-453: "The present study suggests that this relationship is not one-sided, and poly(A)-tail lengths can be regulated by exonucleolysis independently of translation rate and extensively adjusted to changing growth conditions." The paper does not provide enough evidence to suggest a relationship between translation rate and poly(A) tail lengths)

Dear Editor and Reviewers,

We wish to thank the Reviewers for the constructive comments to which we have responded below. In addition to modifications requested by the Reviewers, we have introduced changes to the manuscript due to the necessity to fit the formatting requirements. Sections that have been heavily modified are highlighted in color. In addition, a list of other major modifications, which were not directly requested by the Reviewers, but necessary to fit the formatting requirements, is included below the Reviewer Comment section.

REVIEWER COMMENTS

Reviewer #1 (Remarks to the Author):

In this nice manuscript, the authors take advantage of a state of art direct RNA sequencing technique they developed, using nanopore sequencing, to determine genome wide, and in an unbiased way, the poly(A)-tail length of individual transcripts under various genetic backgrounds and growing conditions.

Obviously, this is only the beginning of the use of this technique and this manuscript somehow sets up the foundations of many further analyses that will be using this approach.

Yet, in this paper, the authors manage to already address a number of questions, which remained mainly unsolved or only analyzed with reporter transcripts, with all the bias that it inevitably introduce.

In particular, they analyze the consequences of the absence of each most relevant exonucleases, deadenylases and poly(A) polymerases. This reveals in particular for the first time the involvement of the Trf5/TRAMP-exosome complex in regulating the poly(A)-tail length of mRNAs. It also settle the long-standing question of the specific role of the two deadenylase complexes, the CCR4-NOT complex and the PAN2/3 complex.

I have only one minor comment. The authors explain that the approach cannot measure the length of poly(A)-tails shorter than ~15 nucleotides, and they state that “The fraction of adenylated mRNA that was recovered did not depend on expression levels or DRS poly(A)-tail length estimation, indicating that factors other than poly(A)-tail length (e.g., secondary structures) also determine recovery efficiency.”

I wonder if this might not rather reflect the fact that important fractions of many mRNAs might be present in an essentially deadenylated or oligo-adenylated form (see in particular Presnyak et al., Cell, 2015; <http://dx.doi.org/10.1016/j.cell.2015.02.029>). It might be important to address this question, either experimentally on a few individual transcripts or may be simply by looking if the efficiency of polyadenylated transcript recovery does not correlate with the fraction of polyA - vs polyA+ mRNAs, as determined by Presnyak et al.

We thank the Reviewer for this important comment which underlines the limitations of our study. A similar concern was also raised by Reviewer 2. To clarify, there are three potential sources of biases in the methodology we employed in our study:

- 1. efficient estimation of poly(A) tail lengths shorter than 10 adenosines was not yet reported (Workman et al., 2019, Krause et al., 2019), and it is not clear whether transcripts with shorter tails are efficiently included in the sequencing library*
- 2. priming by splint ligation on oligo-dT during the library preparation, because of which*

completely deadenylated/non-adenylated RNA species are not included in the library (as in similar (dT)-primed RNAseq libraries)

- 3. the poly(A)⁺ fraction enrichment on oligo-dT resin, used to enhance the yield of DRS sequencing*

Ad1

In the revised version of the manuscript, we show that the DRS chemistry and the Nanopolish algorithm are capable of accurate identification of poly(A) tails of various lengths, including the short ones, which has been demonstrated on a series of artificial spike-ins in the revised Supplementary Fig. 1g (shortest length tested 10 As).

Ad2

Initially, we checked whether there was a significant deadenylated mRNA fraction in RNA samples by re-analyzing our previously published genome-wide transcriptomic data (Tudek et al., 2018; Schmid et al., 2018) where sequencing libraries were prepared with oligo-(dT)-priming with or without prior ribodepletion and in vitro polyadenylation. No major stabilization of any fraction of mRNAs was observed, while mature non-adenylated sn/snoRNAs were greatly up-regulated in the in vitro polyadenylated fraction (Supplemental Fig. 1f in the revised manuscript). This indicated that non-adenylated mRNAs are scarce. We further validated those observations by obtaining a strong correlation of our DRS counts with published datasets prepared with tiling arrays (Gudipati et al., 2012; Pearson $r = 0.7$) or RNA-seq on ribodepleted samples (Presnyak et al., 2015; Pearson $r = 0.71$); (Supplemental Fig. 1b in the revised manuscript). Please note that the group of mRNAs with altered expression in DRS compared to tiling were not identified as differentially expressed in the Presnyak et al. dataset and are all highly expressed intron-containing transcripts that had their expression erroneously estimated in the tiling array data (see Gudipati et al., 2012 and references to material and methods in Xu et al. 2011 doi:10.1038/msb.2011.1). Overall this shows that DRS is largely comparable to non-poly(A) based transcriptomic techniques confirming that the non-adenylated mRNA fraction is most likely a marginal fraction of the coding transcriptome.

Ad3

To estimate the oligoadenylated fraction more precisely, we subjected WT total RNA samples (without enrichment on oligo-(dT)₂₅ resin) to DRS sequencing, which is an equivalent to the preparation of RNA-seq libraries by (dT)-priming. As a result we obtained two biological replicate DRS and one technical repeat, which were averaged for further analysis. It is important to stress that this type of approach is not recommended for the Nanopore technology as it leads to decreased sequencing efficiency due to inefficient adapters ligation and blocking of nanopores by highly structured non-adenylated RNA species (e.g. rRNAs). The DRS on poly(A)⁺ fraction and on total RNA highly correlated with one another in terms of transcripts abundance (Pearson's $r = 0.96$, Figure 1b), again indicating that there was no subgroup of mRNAs enriched for non-adenylated species. However, while mean poly(A) tail lengths strongly correlated (Pearson's $r = 0.72$; Figure 1b), the mean poly(A) tail estimation in the total DRS decreased by an average of 3.5 As, which was due to the capture of an mRNA fraction with poly(A) tails mostly shorter than 10 adenosines (see Supplemental Fig. 1h and Supplemental Figure 1i).

Using filtering based on abundance and change in mean poly(A) tail length estimation in the total versus the poly(A)⁺ fraction we identified 840 coding transcripts having a fraction with short poly(A) tails large enough to weigh heavily on the mean poly(A) tail estimation and on abundance (see Supplemental Table 1 in the revised manuscript). The criteria defined, and validated with a set

of mRNAs tested by qPCR data (see Supplemental Fig. 1j in the revised manuscript), were:

a. decrease in mean poly(A) tail length of at least one standard deviation in each of the four expression groups (which is 1.72, 2.08, 2.76 and 4.15 for mRNAs of high, medium, intermediate, and low expression, respectively - as previously defined in the Methods section and calculated based on the four DRS poly(A)⁺ fraction datasets from WT cells – each composed of at least two biological replicates).

b. mRNA up-regulation by at least 1.2 fold ($\log_2 \rightarrow 0.26$) in the total compared to the poly(A)⁺ sample. We assumed that a decrease in mean poly(A) tail length estimation in total versus poly(A)⁺ that was really due to the existence of a significant oligo-adenylated mRNA fraction would lead to an increase in the abundance of the mRNA. This was validated independently by correlating qPCR data, showing the % recovery of mRNA in the poly(A)⁺ fraction from the total, with changes in mean poly(A) tail length and abundance estimation in the poly(A)⁺ and total RNA DRS datasets (see revised Supplemental Fig. 1j). Consequently, other changes in mean poly(A) tail length estimation are more likely to be related to random variability mostly caused by low read coverage.

Importantly, those results do not heavily influence the main findings of our work regarding change of poly(A) length upon depletion/deletion of exosome or deadenylase complexes components. As the precision of mean poly(A) tail length estimation can be low for mRNAs of minimal abundance, we previously defined the criteria for nuclease sensitivity to be mean poly(A) tail elongation by at least 10 As, regardless of the expression level. This was done to select strong substrates and to reduce the probability of false positives as the variability of most of the mean poly(A) tail length estimations in WT cells do not surpass +/-10 adenosines. The oligo-tailed mRNA fraction lost during poly(A)⁺-fractionation is too small to explain the changes in mean poly(A) tail lengths in *Dis3*-/*Rrp6*-AA, *ccr4* Δ and *pan2* Δ cells, especially in case of deadenylases where no change in mRNA abundance was noted. Accordingly, mRNAs with a large oligoadenylated fraction constitute only 16 % and 19 % of strong deadenylase and exosome targets, respectively.

In the revised version of the manuscript, we consider the limitations of our study once discussing our results.

Gudipati RK, Xu Z, Lebreton A, Séraphin B, Steinmetz LM, Jacquier A, Libri D. Extensive degradation of RNA precursors by the exosome in wild-type cells. *Mol Cell*. 2012 Nov 9;48(3):409-21. doi: 10.1016/j.molcel.2012.08.018. Epub 2012 Sep 20. PMID: 23000176; PMCID: PMC3496076.

Presnyak V, Alhusaini N, Chen YH, Martin S, Morris N, Kline N, Olson S, Weinberg D, Baker KE, Graveley BR, Collier J. Codon optimality is a major determinant of mRNA stability. *Cell*. 2015 Mar 12;160(6):1111-24. doi: 10.1016/j.cell.2015.02.029. PMID: 25768907; PMCID: PMC4359748.

Tudek A, Schmid M, Makaras M, Barrass JD, Beggs JD, Jensen TH. A Nuclear Export Block Triggers the Decay of Newly Synthesized Polyadenylated RNA. *Cell Rep*. 2018 Aug 28;24(9):2457-2467.e7. doi: 10.1016/j.celrep.2018.07.103. PMID: 30157437; PMCID: PMC6130047.

Schmid M, Tudek A, Jensen TH. Simultaneous Measurement of Transcriptional and Post-transcriptional Parameters by 3' End RNA-Seq. *Cell Rep*. 2018 Aug 28;24(9):2468-2478.e4. doi: 10.1016/j.celrep.2018.07.104. PMID: 30157438; PMCID: PMC6130049.

Workman RE, Tang AD, Tang PS, Jain M, Tyson JR, Razaghi R, Zuzarte PC, Gilpatrick T, Payne A, Quick J, Sadowski N, Holmes N, de Jesus JG, Jones KL, Soulette CM, Snutch TP, Loman N, Paten B, Loose M, Simpson JT, Olsen HE, Brooks AN, Akeson M, Timp W. Nanopore native RNA sequencing of a human poly(A) transcriptome. *Nat Methods*. 2019 Dec;16(12):1297-1305. doi: 10.1038/s41592-019-0617-2. Epub 2019 Nov 18. Erratum in: *Nat Methods*. 2020 Jan;17(1):114. PMID: 31740818; PMCID: PMC7768885.

Krause, M., Niazi, A.M., Labun, K., Cleuren, Y.N.T., Müller, F.S., Valen, E., 2019. *tailfindr: alignment-free poly(A) length measurement for Oxford Nanopore RNA and DNA sequencing. RNA* 25, 1229–1241. <https://doi.org/10.1261/rna.071332.119>

Please note that since Supplemental Fig. 1b was updated with correlations from Presnyak et al., 2015 and Gudipati et al., 2008 datasets we felt like correlations with the data from Yassour et al., 2009 and Tudek/Schmid et al., 2018 were redundant and were therefore removed. As a consequence, the following reference was also removed from the list:

Yassour, M. et al. *Ab initio construction of a eukaryotic transcriptome by massively parallel mRNA sequencing. Proc. Natl. Acad. Sci.* 106, 3264–3269 (2009).

Likewise, I don't understand the y-axis in Suppl. Fig. 3C, right panel, since it shows DRS measurements of poly(A)-tail lengths that begin at zero. There might be a scale adjustment to be made here?

There are two reasons for such display of the data. First is the nature of the kernel density function used to determine the shape of a violin plot. In the case of a low number of observations (like for the WT with only a few reads), the density function is inaccurate, which adversely affects the shape of violin plots observed on the figure. Second, DRS estimates poly(A) tail length but does not measure it precisely. If a large fraction of snR13 reads have poly(A) tails of around 10As, then the estimate is most frequently 10, but its outliers could fall anywhere between 0 and 20 As. This has been demonstrated by Workman et al. (2019) on a series of transcripts with fixed poly(A) tail lengths (Fig. 5A in Workman et al., 2019) and by us, in the revised version of this manuscript (please see Supplemental Fig. 1g). We are afraid that a scale adjustment would eliminate this information and only put forward reads with long poly(A) tail, which can be misleading. This is also why we applied a broad cut-off, ± 10 As, for considering a significant change in the length of the poly(A) tail.

To provide more clarity to the data, the violin plots in Supplemental Figure 3c now display single-read information in the form of beeswarm plots.

Workman RE, Tang AD, Tang PS, Jain M, Tyson JR, Razaghi R, Zuzarte PC, Gilpatrick T, Payne A, Quick J, Sadowski N, Holmes N, de Jesus JG, Jones KL, Soulette CM, Snutch TP, Loman N, Paten B, Loose M, Simpson JT, Olsen HE, Brooks AN, Akeson M, Timp W. Nanopore native RNA sequencing of a human poly(A) transcriptome. *Nat Methods.* 2019 Dec;16(12):1297-1305. doi: 10.1038/s41592-019-0617-2. Epub 2019 Nov 18. Erratum in: *Nat Methods.* 2020 Jan;17(1):114. PMID: 31740818; PMCID: PMC7768885.

Reviewer #2 (Remarks to the Author):

In this manuscript, Tudek and colleagues proposed to investigate the role of different decay factors in determining the poly(A) tail length in budding yeast. The authors used direct RNA sequencing, developed by Oxford Nanopore Technology, to assess the distribution of poly(A) tail lengths and transcript abundance. A set of bioinformatic tools were then used to analyze the generated data and interpret all the correlations. Overall, the manuscript contains a rich amount of data and the authors attempt to assess the correlation between poly(A) tail length and mRNA expression from multiple angles. Although the title of this study mentions “polyadenylation landscape”, the manuscript focuses rather on the poly(A)-tails (particularly their length) than the process of polyadenylation itself, which is misleading.

While the second section of the manuscript does focus on polyadenylation by Pap1, we see that the title of the article does not precisely reflect the content. The title has been changed to:

'Global view on the metabolism of RNA poly(A) tails in the yeast Saccharomyces cerevisiae'

My principal and most important criticism of the novelty claims in this manuscript, which center around the fact that authors applied direct RNA sequencing, are undermined by the fact that poly(A) tails shorter than 15 As cannot be detected by this technology. Given that poly(A) tails of yeast mRNA are already quite short, and in absence of validation by other techniques, this rather limits the interpretation of poly(A) tail metabolism as presented in this study.

We agree with the Reviewer that the bias toward longer poly(A) tails can be a clear limitation of our study, though we do not understand why the Reviewer linked it with the novelty issue.

The concern about the usability of DRS for oligoadenylated RNA species was also raised by Reviewer 1. We addressed this issue extensively both experimentally (DRS on total RNA samples - not subjected to oligo-d(T) enrichment and not ribodepleted) and through the re-analysis of existing RNA-seq datasets of total and poly(A) selected RNAs. We have also sequenced a set of spike-ins with pre-defined poly(A) lengths, confirming that our approach is suitable for poly(A) tails equal or longer than 10A. All this is described in detail in our response to Reviewer 1 (see above). Overall, we show that the poly(A)⁺ fraction is a good approximation for the coding transcriptome and that the biases arising from our experimental approach do not diminish the main conclusions of the paper.

The concern regarding the absence of validation is not valid as in the original version of the manuscript, we showed high resolution-northern blots in Fig 3d and Supplemental Fig. 3c, which match very well with DRS (please notice that both techniques measure the lengths of RNA rather than PCR amplified cDNA and, as such, are superior regarding the trustability in comparison to any other technique). Moreover, in the revised version of the manuscript, we further validate the estimations of the poly(A) tail lengths of RNAs by DRS using in vitro transcribed spike-ins.

The novelty issue is addressed in responses to other concerns raised by the Reviewer.

Another major technical concern is that the sequencing of poly(A) tails in the background of depleted deadenylase enzymes in a corresponding strain (Δ ccr4, Δ pop2 and Δ pan2) is a poorly designed experiment. It has been shown previously that Ccr4 is the principal deadenylase in budding yeast and this is confirmed in this study. However, depleting Pop2 leads to the concomitant depletion of Ccr4 and, consequently, disrupting the deadenylase activity of the Ccr4-Not complex as a whole. The authors somehow failed to take this into account and the data is thus very difficult to interpret.

In the revised version of the manuscript, we only compare the effects of deletion of CCR4 and PAN2 (both are the main catalytic subunits), which should result in near-to or complete inactivation of deadenylation by their respective complexes. We, however, wish to point out that our results concerning Pop2 function in deadenylation are in line with the limited genome-wide data available online (see Balagopal et al. preprint → figure pasted below). The concerns raised by the Reviewer circle around the function of the NOT core and Pop2 in relation to Ccr4 catalytic activity, which is a more complex issue, and we see how our cursory approach is insufficient.

Vidya Balagopal, Mohan Bolisetty, Najwa Al Husaini, Jeff Collier, Brenton R. Graveley. Ccr4 and Pop2 control poly(A) tail length in Saccharomyces cerevisiae. bioRxiv 140202; doi: <https://doi.org/10.1101/140202>

Data from Balagopal et al. (2017), preprint:

Figure 4

The majority of sections of this manuscript should be re-written and presented in a sequential and coherent manner. The reader might find some difficulties in keeping up with the narration of results and the underlying conclusions. The Introduction is an accumulation of literature on RNA polyadenylation and degradation factors in yeast without any clear axis of interest related to the aim of the study and in a confusing way. There is no linkage between paragraphs and there are abrupt transitions from one mechanism to another. One example from the Results is already the first paragraph (lines 119-134). The text is focused on a supplementary figure which is supposed to support the key result presented in Figure 1, but less description is applied to Figure 1 itself. There are other instances of confusing and meandering writing.

The structure of all sections of the manuscript has been improved.

Specific concerns:

Lines 75-7: “During its cytoplasmic lifespan, the poly(A) tail is gradually shortened by two main deadenylases PAN2/3 and CCR4-NOT”.

This statement is misleading in many ways, not least because the work by Narry Kim laboratory had provided compelling evidence that it is the CCR4-NOT that acts as the principal factor responsible for poly(A) tail control in the cytoplasm.

*It is important to stress out that the Narry Kim group focused on the study of deadenylation in various human cell lines using reporter mRNAs, TAIL-seq, and bulk poly(A) assay. While many mechanisms might be conserved from yeast to humans, they cannot be simply extrapolated to the yeast system. In yeast studies on poly(A) tail length control are mostly limited to reporter transcripts. Our work is focused only on yeast *Saccharomyces cerevisiae*, and we wished to introduce the potential main players in deadenylation in an unbiased way.*

There is no evidence that it does so in concert with PAN2/3, which this statement implies.

We are quite surprised by this statement. Considering the works from Narry Kim and Ann-Bin Shyu groups alone (which again focused on human cell lines; Chang et al., 2014; Yi et al.,

2018; Yamashita et al., 2005), evidence supporting the functional cooperation of both deadenylases at certain steps of deadenylation exists (quote from Yi et al., 2018 regarding the bulk poly(A) assay: 'Strong accumulation of ~150 nt fragments in CAF1-depleted cells indicated the CNOT complex mainly acts on ~150 nt poly(A) tails for most transcripts. (...) However, the PAN2/3 depletion slightly but reproducibly increased the long (>150 nt) poly(A)s.' (...) 'Thus, while PAN2/3 trims long poly(A) tails to a fixed length (~150 nt), which is consistent with the biphasic model (Figure S2E), this first phase is unlikely to be a critical prerequisite for deadenylation and decay.').

Those conclusions were sadly drawn from bulk poly(A) tail assays because TAIL-seq is not capable to efficiently detect poly(A) tails longer than 128 As, limiting the use of this method to draw general conclusions. Bulk poly(A) tail assays have the down-fall of over-representing the effects present on a handful of highly expressed mRNAs. Moreover, as widely acknowledged, siRNA-depletions in higher eukaryotes are seldom complete and have different time onsets. Thus taking those results as a definitive proof is not admissible. They instead invite further investigation (by applying the CRISP-Cas9-mediated deletion or conditional protein degradation by Auxin Inducible Degron).

The data collected for yeast also indicates functional complementation of CCR4-NOT and PAN2/3. An early model for deadenylation-dependent mRNA decay was proposed by Muhlard et al. (1994), based on the decay of reporter MFA2 gene in *S. cerevisiae*. The authors proposed that deadenylation occurs in phases. Initial, slow deadenylation later accelerates and leads eventually to a substantial shortening of the poly(A) tail, which results in decapping and the subsequent rapid decay of the transcript from the 5' end. Later, structural data obtained by Wolf et al., (2014) and Schafer et al., (2019), showed that yeast PAN2/3 is recruited by poly(A) RNA binding protein to a surface created by two Pab1 molecules bound to a poly(A) RNA sequence. This data could explain the postulated preferential role of PAN2/3 in the initial deadenylation phase since poly(A) tail can accommodate several Pab1 molecules. Recently, the CCR4-NOT complex activity was also shown to be regulated by Pab1 *in vitro*, and we include the relevant literature in the revised version of the manuscript. Webster et al., (2018) show that in *S. pombe* and *S. cerevisiae*, Ccr4 degrades Pab1-bound poly(A) transcripts and that it is important for removal of the last 22 As, both *in vivo* and *in vitro*. Collectively, this *in vitro* data is also in line with the bi-phasic deadenylation model. However, there is a modification that is also apparent in our *in vivo* DRS datasets. CCR4-NOT is capable of degrading poly(A) tails potentially containing multiple Pab1 molecules and thus also functions to some extent in the first deadenylation phase.

Those works and our data do not infer a direct interaction between PAN2/3 and CCR4-NOT. Neither of the works argues the quantitative importance of one deadenylase complex over the other for cellular metabolism. They rather show that each has a different substrate specificity and mechanism of action. This is later underlined in the Discussion where works showing that CCR4-NOT is also recruited to mRNAs by adapter proteins are cited.

The work cited based on *in vivo* reporters has largely been replaced by a more comprehensive view based on measurements of poly(A) tail lengths in cells of the transcriptome.

In yeast, the majority of the peer-reviewed, published data regarding deadenylation are based on reporter mRNAs and *in vitro* deadenylation assays. This is why our work provides substantial novelty, not only to the yeast community but also in general, as juxtaposed to the wealth of human-based research, it points out the changes in the deadenylation mechanism between species.

Chang H, Lim J, Ha M, Kim VN. TAIL-seq: genome-wide determination of poly(A) tail length and 3' end modifications. *Mol Cell*. 2014 Mar 20;53(6):1044-52. doi: 10.1016/j.molcel.2014.02.007. Epub 2014 Feb 27. PMID: 24582499.

Muhrad D, Decker CJ, Parker R. Deadenylation of the unstable mRNA encoded by the yeast MFA2 gene leads to decapping followed by 5'→3' digestion of the transcript. *Genes Dev.* 1994 Apr 1;8(7):855-66. doi: 10.1101/gad.8.7.855. PMID: 7926773.

Yamashita A, Chang TC, Yamashita Y, Zhu W, Zhong Z, Chen CY, Shyu AB. Concerted action of poly(A) nucleases and decapping enzyme in mammalian mRNA turnover. *Nat Struct Mol Biol.* 2005 Dec;12(12):1054-63. doi: 10.1038/nsmb1016. Epub 2005 Nov 13. PMID: 16284618.

Yi H, Park J, Ha M, Lim J, Chang H, Kim VN. PABP Cooperates with the CCR4-NOT Complex to Promote mRNA Deadenylation and Block Precocious Decay. *Mol Cell.* 2018 Jun 21;70(6):1081-1088.e5. doi: 10.1016/j.molcel.2018.05.009. PMID: 29932901.

Schäfer IB, Yamashita M, Schuller JM, Schüssler S, Reichelt P, Strauss M, Conti E. Molecular Basis for poly(A) RNP Architecture and Recognition by the Pan2-Pan3 Deadenylase. *Cell.* 2019 May 30;177(6):1619-1631.e21. doi: 10.1016/j.cell.2019.04.013. Epub 2019 May 16. PMID: 31104843; PMCID: PMC6547884.

Wolf, J. et al. Structural basis for Pan3 binding to Pan2 and its function in mRNA recruitment and deadenylation. *EMBO J.* 33, 1514–1526 (2014).

Reference added to the list:

Webster MW, Chen YH, Stowell JAW, Alhusaini N, Sweet T, Graveley BR, Collier J, Passmore LA. mRNA Deadenylation Is Coupled to Translation Rates by the Differential Activities of Ccr4-Not Nucleases. *Mol Cell.* 2018 Jun 21;70(6):1089-1100.e8. doi: 10.1016/j.molcel.2018.05.033. PMID: 29932902; PMCID: PMC6024076.

Lines 105-5: “We found that mRNA poly(A)-tail lengths are generally not correlated with transcript translation efficiency...”

This is also misleading because there are no measurements of translational efficiency described in this study, such as those that may be obtained by ribosome profiling combined with deep sequencing.

Our study aimed at a global description of poly(A) length regulation in budding yeast. We provide over 50 sequencing runs to describe various aspects of mRNA poly(A) tail metabolism mostly related to the activity of the main 3'-5'-exonucleases. The hypothesis that poly(A) tail length regulation is coupled to translation is a longstanding question, but is not central to our biological questions. We have thus taken advantage of published translation rate estimates to approximate an answer to this question.

The work by Siwiak and Zielenkiewicz (2010), included previously in the manuscript, estimates the number of protein molecules produced by each mRNA during its lifespan for nearly the entire coding transcriptome based on a ribosome profiling dataset from Ingolia et al., (2009) and additional mathematical modeling. A lack of steady-state poly(A) tail length correlation with translation efficiency is consistent with previous reports from Subtelny et al., 2014 and was thus not investigated any further. To make those data clearer for readers willing to evaluate a raw estimate of translation efficiencies (ribosome occupancy) such data were also included in the revised manuscript in Figure 1f bottom panels.

*Ingolia NT, Ghaemmaghami S, Newman JR, Weissman JS. Genome-wide analysis in vivo of translation with nucleotide resolution using ribosome profiling. *Science.* 2009 Apr 10;324(5924):218-23. doi: 10.1126/science.1168978. Epub 2009 Feb 12. PMID: 19213877; PMCID: PMC2746483.*

Lines 142-2: “Such heterogeneity of poly(A)-tail lengths increased gradually with decreasing expression”.

This statement, in its most direct interpretation, is simply not supported by the data presented because the estimates of mRNA levels are used as a proxy for mRNA expression and translation. This weakens the analysis, as there is a substantial number of mRNAs that can exist in a

translationally silenced state. The authors should at the very least acknowledge this fact before drawing conclusions.

*This sentence does not refer to translation efficiency but simply to the level of mRNA. We admit that the use of the word 'expression' was a poor choice as it is a sum of both transcription and RNA half-life we have thus changed it to **mRNA abundance** throughout the manuscript.*

Lines 144-5: “We instead suggest that individual RNAs undergo different poly(A)-tail length control.”

There is indeed a heterogeneity in poly(A) tail lengths as determined by the experimental approach the authors have taken. This is neither a surprising observation nor a novel one. There is no evidence in this study for differences in regulation.

This speculation was removed.

Lines 149-150: “...rather than poly(A)-tail length, mRNA abundance is a function of transcription and synthesis rates.”

What the authors observe, in fact, that there is no correlation between poly(A) tail lengths and mRNA half-life. The suggestion about the determinant of mRNA abundance remains purely speculative.

Full quote: 'Comparisons of our DRS results with existing datasets revealed that rather than poly(A)-tail length, mRNA abundance is a function of transcription and synthesis rates (Fig. 1b, Supplemental Fig. 1h).'

The Table below sums up the Pearson r coefficients given for scatterplots in revised Fig. 1d and Supplemental Fig. 1k, which remain unchanged compared to the previous manuscript version.

Estimate of mRNA synthesis rate vs DRS	DRS mRNA abundance	DRS mRNA mean poly(A) tail length
mRNA synthesis rate (Sun et al., 2012)	0.71	0.38
mRNA synthesis rate (Neymotin et al., 2014)	0.62	0.34
RNA polymerase gene occupancy (Pelechano et al., 2010)	0.61	0.34

We conclude that the correlation between mRNA production rate (estimated either by metabolic labelling or the level of RNA polymerase on a gene) is stronger between mRNA abundance than mean poly(A) tail length estimate. This data supports the sentence fully in the main text, but we agree that the phrasing might have been confusing. Thus, it was replaced with: “Transcription level estimates best defined DRS mRNA abundance but were poorly correlated with mean poly(A) tail lengths.”

The original sentence does not refer to mRNA half-lives in any way, but the Reviewer is right that we do not see any strong correlation between mRNA half-lives and DRS estimated mean poly(A) tail lengths as shown later on in revised Fig. 1e ($r = [-0.39 \text{ to } -0.22]$).

Lines 152-5: there is a confusion of concepts between the “poly(A)-tail of short length (30-40 nucleotides)” and “reduction of poly(A)-tail length (deadenylation)”.

To state “This indicates that a substantial reduction of poly(A)-tail length might not necessarily lead to rapid decay” might be an erroneous assumption because no data shown until this point is assessing decay rate as a function of the deadenylation process. The RNAs are still at the steady-state. Furthermore, in the same paper used as reference (Eisen et al, Mol Cell 2020), the rate (or “speed”) of deadenylation is a function of the “degree” because the speed increases below a specific degree of deadenylation.

The sentence only highlighted the stability of highly expressed mRNAs, which at steady state bear short poly(A) tails. Since it is merely speculation on the possible mechanism, it has been removed from the revised version.

The reference to the work by Eisen et al., (2020) has also been removed.

Lines 157-8: “The number of protein molecules per cell correlated strongly with mRNA abundance and anticorrelated with mean poly(A)-tail length²⁷ (Fig. 1d), which can be attributed to co-translational mRNA deadenylation.”

The fact that the length of the poly(A)-tail is anticorrelated with translation efficiency has already been shown by others. The authors speculate that the same trend is observed in the budding yeast. I use the term “speculate” because there is no direct experimental evidence in this paper to make claims regarding any kind of correlation/anti-correlation of the number of protein molecules per length of the mRNA poly(A) tail. The cited papers do not provide clear distinct direct evidence for this either. The text nor the figure legend provides no description of how the authors arrived at this conclusion.

Since former Figure 1d (in the revised version Fig. 1f) top-left and top-right panels are the ones with the strongest Pearson r factor, we assume that those are the ones the Reviewer relates to in this comment. However, those panels and the supporting text do not have anything to do with translation efficiency measured by ribosome-profiling, and issues related to translation efficiency have been addressed in response to comments related to the work of Siwiak and Zielenkiewicz (2010) above and revised Figure 1f middle and bottom panels.

Please note that the revised Figure 1f top-left and top-right panel does not relate to translation efficiency but shows a scatterplot of the mean poly(A) tail lengths and mRNA abundance measured with DRS, obtained in our laboratory, and the number of protein molecules derived from the work of Ho and co-workers (2019). We and Ho et al. (2019) show that the mRNA abundance is the best determinant of the number of protein molecules in the cell at a steady state. Ho et al. utilized 21 published datasets to derive a unified model of the number of each protein molecule per cell. Those datasets were obtained by either mass-spectrometry, GFP-microscopy, GFP-flow cytometry or TAP-immunoblotting from cells grown in both minimal and rich media, mostly at mid-log phase. Those are conditions that we analyze in our manuscript. The work of Ho et al. (2018) is widely cited (over 100 citations), including by the main yeast data repository yeastgenome.org and since it is based on the work of many others, we find it is likely to reflect best the true average biological situation at mid-log growth phase. We believe that any other dataset that we would obtain independently would not add novelty to the abundance of data available and would bear much more experimental bias.

We again stress that our cursory approach to the translation-protein output-poly(A) tail length connection stems from the fact that we focused our work on the activity of the main cellular nucleases and their co-factors. While acknowledging this, please note that our conclusions are by no means controversial in the light of the literature in the field.

Ho B, Baryshnikova A, Brown GW. Unification of Protein Abundance Datasets Yields a Quantitative *Saccharomyces cerevisiae* Proteome. *Cell Syst.* 2018 Feb 28;6(2):192-205.e3. doi: 10.1016/j.cels.2017.12.004. Epub 2018 Jan 17. PMID: 29361465.

Siwiak M, Zielenkiewicz P. A comprehensive, quantitative, and genome-wide model of translation. *PLoS Comput Biol.* 2010 Jul 29;6(7):e1000865. doi: 10.1371/journal.pcbi.1000865. PMID: 20686685; PMCID: PMC2912337.

Lines 199-203: the difference between pap1-1 rrp6 Δ and the other two (pap1-1 & pap1-1 trf4 Δ) is clearly significant (almost 50% recovery). It is not at all clear how the authors assessed the significance between averages of 6.86 and 7.66. Drawing this conclusion “Trf4 and Trf5 proteins can compensate for its activity” is speculative.

The Reviewers doubts are most likely related to a misunderstanding of the data. We reformulated the conclusion for this section for more clarity. Trf4 is the closest Rrp6 co-factor, responsible for stimulation of Rrp6 exonucleolysis in a polyadenylation-dependent and independent manner (Callahan et al., 2010; Schuch et al., 2014). In turn, Trf4 polyadenylation is only clearly detected in rrp6 Δ background (Carneiro et al., 2007). The full quote:

“ mRNA poly(A) tails were shortened by an average of 6.86, 3.73, and 7.66 adenosines in the pap1-1, pap1-1 rrp6 Δ , and pap1-1 trf4 Δ backgrounds, respectively” - Lines 204-205

Thus deletion of RRP6, but not TRF4, in pap1-1 background, results in partial rescue of growth, and mRNA abundance and poly(A) tail length. The lack of rescue in the pap1-1 trf4 Δ background indicates that the mechanism of the rescue observed in pap1-1 rrp6 Δ is not purely related to the impairment of Rrp6 decay directed towards Pap1-produced poly(A) tails but due to stabilization of Trf4/5-borne poly(A) tails. In other terms Trf4/5 polyadenylates mRNA 3'ends, but those tails are trimmed by Rrp6 in wild-type cells.

The effects are clear both on the estimation of mean poly(A) tail length (for which we have now included appropriate statistics in the text and on the plots) and in single gene examples. The panels displaying the latter have been updated for more clarity (see Fig. 2h and Supplemental Fig. 2f) and now include not only poly(A) tail length distribution but also information on the number of reads and library-size normalized counts.*

**Through-out the figures the statistical significance of mean mRNA poly(A) tail length change was assessed using the Kruskal-Wallis test with Dunn test for pairwise comparisons.*

Callahan KP, Butler JS. TRAMP complex enhances RNA degradation by the nuclear exosome component Rrp6. *J Biol Chem.* 2010 Feb 5;285(6):3540-7. doi: 10.1074/jbc.M109.058396. Epub 2009 Dec 2. PMID: 19955569; PMCID: PMC2823493.

Schuch B, Feigenbutz M, Makino DL, Falk S, Basquin C, Mitchell P, Conti E. The exosome-binding factors Rrp6 and Rrp47 form a composite surface for recruiting the Mtr4 helicase. *EMBO J.* 2014 Dec 1;33(23):2829-46. doi: 10.15252/embj.201488757. Epub 2014 Oct 15. PMID: 25319414; PMCID: PMC4282559.

Carneiro T, Carvalho C, Braga J, Rino J, Milligan L, Tollervey D, Carmo-Fonseca M. Depletion of the yeast nuclear exosome subunit Rrp6 results in accumulation of polyadenylated RNAs in a discrete domain within the nucleolus. *Mol Cell Biol.* 2007 Jun;27(11):4157-65. doi: 10.1128/MCB.00120-07. Epub 2007 Apr 2. PMID: 17403903; PMCID: PMC1900014. → ADDED TO THE REVISED MANUSCRIPT

Lines 208-211 present another confusion of concepts. Deficient adenylation is one process, and deadenylation is another.

The Reviewer correctly pointed out that instead of 'deadenylation' the term 'shortening of poly(A) tail' should be employed. This has been corrected in this paragraph and elsewhere if applicable. Furthermore, to improve clarity, the entire paragraph has been simplified.

This section begins with reference to a publication by Burkard and Butler (2000), that shows strong down-regulation of a few reporter mRNAs at restrictive temperature in the pap1-1 strain and suggested that deficient adenylation results in rapid decay of the mRNA. We show that the pap1-1 strain is indeed deficient in processive adenylation in vivo, but surprisingly the degree of the defect does not strongly correlate with mRNA down-regulation, even though an overall loss of poly(A)+ RNAs in pap1-1 and pap1-1 trf4Δ compared to both WT and pap1-1 rrp6Δ is evident and conceptually confirms the Burkard and Butler (2000) report. This is shown in scatterplots that tie mRNA abundance and mean poly(A) tail length changes in Fig. 2e,f,g and in the revised panels showing single gene examples in Fig. 2h and Supplemental Fig. 2f, which, in addition to poly(A) tail distribution, now also include the number of reads and library-size normalized counts. We currently have no explanation for this observation and we thus suggest in the revised version that some transcripts are polyadenylated normally in the pap1-1 and undergo different poly(A) tail length regulation in the nucleus or cytoplasm.

When describing a result, it would be helpful to provide a reference to the details in the figures: “highly expressed mRNAs in these mutant strains were virtually identical to the WT control. In contrast, mRNAs that had alterations of mean poly(A)-tail lengths exhibited alterations of poly(A)-tail patterns (Fig. 2h, Supplemental Fig. 2f)” it is not at all not clear what is what.

Indeed the panel names were missing in the main Figure 2. We have corrected this and improved the main text, as explained in the two responses above.

Lines 300-1: “ A large proportion of mRNAs was upregulated in Dis3- and Dis3/Rrp6-depleted cells” is quite misleading because log₂ mRNA abundance in the bottom panel of Fig 4b does not show a significant change in mRNA levels in depleted cells.

Scatterplots in Fig. 4d show that only a fraction of mRNAs in each strain is upregulated by more than 2-fold (transcripts with log₂ mRNA abundance change more than 1 → on the right to the dashed line). Taking this under consideration, the blue series designating all mRNAs in the violin plot in Fig. 4b might not display a strong change in abundance of all mRNAs (blue series), but the change is nonetheless visible (see graph below). In contrast, mRNAs of high expression (red dots in Fig. 4d and red violins in Fig. 4b, see below) do not substantially change in abundance. Horizontal lines were inserted throughout the manuscript figures to facilitate comparison between strains.

Line 385: “This extensive regulation suggests poly(A)-tail length has, next to the adjustment of mRNA levels, an important impact on shaping protein expression.” Leaving aside the fact that proteins are not expressed but the genes are, it seems that with this statement the authors appear to contradict themselves on line 149: “Comparisons of our DRS results with existing datasets revealed that rather than poly(A)-tail length, mRNA abundance is a function of transcription and synthesis rates”.

We, similarly to Subtelny et al., (2014), do not see a correlation between poly(A) tail length and translation efficiency measured by ribosomal-profiling at steady-state growth conditions in yeast. However, as Subtelny et al. (2014) show such link can be transient in Xenopus and zebrafish embryonic cells. With the statement quoted, we thus speculated that mRNA poly(A) tail length regulation in yeast cells could influence translation during changes in growth conditions. We see now how this was poorly phrased and rather far-fetched. This sentence was thus removed in the revised version.

Subtelny AO, Eichhorn SW, Chen GR, Sive H, Bartel DP. Poly(A)-tail profiling reveals an embryonic switch in translational control. Nature. 2014 Apr 3;508(7494):66-71. doi: 10.1038/nature13007. Epub 2014 Jan 29. PMID: 24476825; PMCID: PMC4086860.

The general feeling one gets from reading this manuscript is that the principal novelty lies in the application of new and exciting methodology to sequence mRNAs. It is not without its limitations and, unfortunately, this manuscript does not provide any new mechanistic insights into mRNA regulation. Apart from significant issues with the design and interpretation of the experiments as well as a general lack of novelty, the manuscript is poorly worded in quite a few places and contains a large number of typos, grammar, and syntax errors. I would encourage the authors to reassess the suitability of this work, in its present form, for publication in any venue.

DRS is, as any other method, not without its limitation, but it is a significant improvement over TAILseq (Chang et al., 2014) and PALseq (Subtelny et al., 2014). Both are Illumina-based and tie poly(A) tail lengths to the last few bases at the RNA 3'end. In contrast, by producing long reads, DRS can distinguish between mRNA isoforms (alternative cleavage sites and splicing isoforms). In addition, TAILseq cannot reliably measure poly(A) tails longer than 128 adenosines, which introduces a considerable bias, especially for higher eukaryotes, where poly(A) tails are longer; but also for yeast as exemplified by our study of de novo polyadenylation, which established the limit of Pap1 activity at 200 adenosines.

We strongly disagree with the Reviewer concerning the lack of biological novelty in our study. We show for the first time that the exosome and, in particular, Dis3 is widely implicated in the regulation of mRNA poly(A) tail lengths in yeast. Since this activity is most likely limited to the nucleus, this might have an impact on mRNA stability and efficiency of export, which opens new lines of future research. Furthermore, we show Trf4/5 polyA-polymerases activity, under specific circumstances, can target mRNA for polyadenylation. This might be important for wild-type cells in certain situations, such as meiosis, when Rrp6 levels are decreased (Lardenois et al., 2010). Finally, we show different substrate specificity for CCR4-NOT and PAN2/3 complexes genome-wide, which sheds new light on the mechanism of the first deadenylation phase, at least in the yeast system. In conclusion, we are convinced that the publication provides novelty both in regards to the technical approach and the biological meaning of the results and would be of interest to all researchers working on RNA metabolism.

Not being English native speakers, we can only apologize for typos, grammar, and syntax

errors. The revised version has been carefully checked for such mistakes.

Chang H., Lim J., Ha M., Kim V.N. (2014). TAIL-seq: genome-wide determination of poly(A) tail length and 3' end modifications. *Mol. Cell.* 53(6):1044-52. doi: 10.1016/j.molcel.2014.02.007.

Lardenois A, Liu Y, Walther T, Chalmel F, Evrard B, Granovskaia M, Chu A, Davis RW, Steinmetz LM, Primig M. Execution of the meiotic noncoding RNA expression program and the onset of gametogenesis in yeast require the conserved exosome subunit Rrp6. *Proc Natl Acad Sci U S A.* 2011 Jan 18;108(3):1058-63. doi: 10.1073/pnas.1016459108. Epub 2010 Dec 13. PMID: 21149693; PMCID: PMC3024698.

Subtelny AO, Eichhorn SW, Chen GR, Sive H, Bartel DP. Poly(A)-tail profiling reveals an embryonic switch in translational control. *Nature.* 2014 Apr 3;508(7494):66-71. doi: 10.1038/nature13007. Epub 2014 Jan 29. PMID: 24476825; PMCID: PMC4086860.

Reviewer #3 (Remarks to the Author):

In this work, “Comprehensive picture of the RNA polyadenylation landscape in the yeast *Saccharomyces cerevisiae*”, the authors investigated the transcriptome-wide changes of poly(A) tail lengths and transcript abundance under various growth conditions and selected mutants of *S. cerevisiae*. The authors use Oxford Nanopore Direct-RNA sequencing (DRS) to investigate poly(A) length and transcript abundance independent of possible amplification artefacts. Through their data they demonstrate that poly(A)-tail length is not directly coupled to transcript abundance or translation regulation, and dissect the mechanisms of poly(A)-tail maturation in the nucleus and cytoplasm of *S. cerevisiae* through analysing relevant enzyme knockout mutants.

The approach is suitable to demonstrate the interplay of various poly(A) processing factors in biological systems, and to show correlative relationships to RNA abundance. The use of Oxford Nanopore Direct RNA sequencing enables deep insights into transcriptomics regulations. While poly(A) measurements with this technology may have somewhat low precision and suffer from low count numbers, they allow transcriptome-wide analysis. The outlined data analysis seems reproducible, and the data is made accessible. With the power of *S. cerevisiae* knockouts, the authors shed light on the complex interplay of poly(A) processing machinery and the outcome of transcript abundance. It is thus a valuable addition to the field of RNA biology.

The conclusions drawn appear to be robust and supported by the presented data. In general the paper is well-written and well structured. The experiments follow logically from each other, and together support the conclusions of the article.

A possible exception is the claim that the maximum poly(A) length is around 200. The data presented for this claim as a biological limit is somewhat weak and should be supported by more than single-gene violin plots ending at 200. We suggest the authors provide values for transcriptome-wide maximal measured poly(A) tails and investigate possible alternative explanations for these measurements. Do all transcripts harbor these lengths, is it specific transcripts? Do these measurements happen often enough to exclude technical artifacts? Are there alternative methods that could verify a maximum poly(A) length?

We agree with the Reviewer that we haven't shown strong evidence for such a limit. Therefore, in the revised version, we show more single gene examples for heat-induced transcripts in both mft1Δ (49 mRNAs; revised Supplemental Figure 2a) and Mex67-AA cells (39 mRNAs; revised Figure 2a). Those revised plots are composed of two panels. A beeswarm plot in the upper panel displays poly(A) tail estimates for single reads and outlines the 99.5 % quantile and mean for the transcripts (95.5 % quantile in dashed line: 198 for Mex67-AA and 163 for mft1Δ compared to approx. 140 in control cells and mean in solid line: 50 As for both mutants). Only transcripts with a large number of reads are displayed. In addition, revised Fig. 2b shows an empirical cumulative

distribution function plot for transcripts from Fig. 2a (Mex67-AA and control strains). The bottom panels of Figure 2a and Supplemental Figure 2a show the sum of library-size normalized counts for each transcript to prove that the majority of the mRNAs taken for the analysis are up-regulated during heat-shock and thus newly made. Please note that most mRNAs up-regulated in wild-type cells during heat shock are degraded in export mutants, while many of those that escape decay bear longer poly(A) tails, as was shown before (Rougemaille et al., 2007; Tudek et al., 2018). The Mex67-AA depletion and heat shock were performed in minimal media, where mean poly(A) tail lengths were shorter than in rich media (see Figure 6 and Supplemental 6 and related text), in which the *mftΔ* and control strain were grown. Despite this, the Mex67 depletion results in an accumulation of heat-induced transcripts with longer poly(A) tails. This might be due to differential Pap1 activity in minimal media but alternatively, and more likely in our opinion, to the export block not being as severe in THO-TREX mutants in general (Tudek et al., 2018). Overall we uphold our 200 adenosine estimation of the maximum de novo polyA-tail length in yeast and set the mean poly(A) tail length of newly made-transcripts at 50 As.

Consequently, the former Figure 2a and the supporting text have been removed as they were now redundant.

Rougemaille M, Gudipati RK, Olesen JR, Thomsen R, Seraphin B, Libri D, Jensen TH. Dissecting mechanisms of nuclear mRNA surveillance in THO/sub2 complex mutants. *EMBO J.* 2007 May 2;26(9):2317-26. doi: 10.1038/sj.emboj.7601669. Epub 2007 Apr 5. PMID: 17410208; PMCID: PMC1864968.

Tudek A, Schmid M, Makaras M, Barrass JD, Beggs JD, Jensen TH. A Nuclear Export Block Triggers the Decay of Newly Synthesized Polyadenylated RNA. *Cell Rep.* 2018 Aug 28;24(9):2457-2467.e7. doi: 10.1016/j.celrep.2018.07.103. PMID: 30157437; PMCID: PMC6130047.

List of Suggestions:

Major:

Line 166: “with the limit of 200 adenosines”

And Line 179-182: “regardless of growth conditions, the polyadenylation limit was around 200 adenosines ... (Fig 2b, Supplemental Fig. 2a).”

Figure 2b shows only one gene and violin plots under various conditions. These plots have an upper limit of 200, but it is unclear whether this is the limit of the actual underlying data. Supplemental Figure 2a. Shows similar type of plots for three other chosen genes. From these graphs it is not clear that 200 is the maximally measured poly(A) length throughout the whole datasets. It is furthermore not possible to judge whether 200 is an arbitrarily chosen upper limit in plotting, a technical limit of the method (both bioinformatically and/or technically through sequencing) or really stemming from biological information. To support this claim, further data should be provided, or the claim has to be weakened to not be confused with a biological fact.

As explained above the revised Figures 2a and Supplemental Fig. 2a now show more single gene examples in the form of a beeswarm plots. As stated previously, the majority of reads are no longer than 200 adenosines. This is not due to a detection limit of DRS, as poly(A) tails in higher eukaryotes measured by DRS surpass this limit (Bilska et al., 2020, Gewartowska et al., 2021).

Bilska A, Kusio-Kobińska M, Krawczyk PS, Gewartowska O, Tarkowski B, Kobyłecki K, Nowis D, Golab J, Gruchota J, Borsuk E, Dziembowski A, Mroczek S. Immunoglobulin expression and the humoral immune response is regulated by the non-canonical poly(A) polymerase TENT5C. *Nat Commun.* 2020 Apr 27;11(1):2032. doi: 10.1038/s41467-020-15835-3. PMID: 32341344; PMCID: PMC7184606.

Gewartowska O, Aranaz-Novaliches G, Krawczyk PS, Mroczek S, Kusio-Kobińska M, Tarkowski B, Spoutil F, Benada O, Kofroňová O, Szwedziak P, Cysewski D, Gruchota J, Szpila M, Chlebowski A, Sedlacek R, Prochazka J, Dziembowski A. Cytoplasmic polyadenylation by TENT5A is required for proper bone formation. *Cell Rep.* 2021 Apr 20;35(3):109015. doi: 10.1016/j.celrep.2021.109015. PMID: 33882302.

Minor:

Violin plots of poly(A) tail length of single transcripts should be supplemented with the number of measurements that lead to that plot. The shape of violin plots can be significantly influenced by the amount of datapoints contributing to it. The reader cannot judge whether the shape is based from sufficient measurements.

All violin plots of single transcripts throughout the manuscript have been supplemented with the number of reads, as requested by the Reviewer.

Figure 3c has unreadable small font in the legend

This has been corrected as requested in the revised version.

Figure 3d: The northern plot quantification graph is not normalised to overall intensity. To be able to compare to the density plot of Nanopore data, each line should sum up to 100%. Instead, at the moment the blue line is above the red line at all times suggesting a higher general intensity level and precluding direct comparison

In the revised manuscript, the Northern blot intensity has been normalized to the mean intensity of the region, which corresponds to the main band on the Northern blot image. This better relates to DRS quantification methodology.

Figure 6b: The Figure description uses the words “induced” and “repressed” even though only transcript levels in correlation to posttranscriptional regulation were measured. The words are misleading as they suggest active induction or repression of transcription. Instead, “down-regulation/reduced” or “upregulation/increased” should be used.

This has been corrected.

Line 40: Cleavage and Polyadenylation Factor should be abbreviated to (CPF) not (CFP)

This has been corrected.

Line 43: “facilitates” should be “facilitate”

This has been corrected.

Line 385-387: “This extensive regulation suggests poly(A)-tail length has, next to the adjustment of mRNA levels, an important impact on shaping protein expression.” The article does not provide any data supporting the claim that extensive poly(A) tail regulation has an impact on shaping protein expression. Actually, the initial claims of the paper rather support the notion that poly(A) length and translation are not coupled in a way that clearly predicts an outcome of one another.

*The exact same issue was raised by Reviewer 2: Similarly to Subtelny et al., (2014), we do not see a correlation between poly(A) tail length and translation efficiency measured by ribosomal-profiling at steady-state growth conditions in yeast. However, as Subtelny et al. (2014) show, such link can be transient in *Xenopus* and zebrafish embryonic cells. With the statement quoted, we thus speculated that poly(A) tail lengths in yeast cells could influence translation during changes in growth conditions. We see now how this was poorly phrased and rather far fetched. This sentence*

was thus removed in the revised version.

Subtelny AO, Eichhorn SW, Chen GR, Sive H, Bartel DP. Poly(A)-tail profiling reveals an embryonic switch in translational control. Nature. 2014 Apr 3;508(7494):66-71. doi: 10.1038/nature13007. Epub 2014 Jan 29. PMID: 24476825; PMCID: PMC4086860.

Line 451-453: “The present study suggests that this relationship is not one-sided, and poly(A)-tail lengths can be regulated by exonucleolysis independently of translation rate and extensively adjusted to changing growth conditions.” The paper does not provide enough evidence to suggest a relationship between translation rate and poly(A) tail lengths)

As the paragraph is purely speculative, it has been removed.

OTHER MANUSCRIPT MODIFICATIONS

Both the Introduction and the Discussion have been shortened to fit the journal's overall maximum number of words. The main message remains unaltered unless otherwise criticized by the Reviewers, in which case the nature of the modification is explained in the responses above.

The graphs (revised Supplementary Figures: 1a, 2b, 3a and 4a) displaying reverse-transcription quantitative PCR results have been modified to adhere to the journal's formatting requirements. Error bars, previously showing deviation from the mean, have been replaced with single dots showing values used to calculate the mean.

- Supplemental Table 1 listing up-regulated and down-regulated mRNAs following the heat shock has been transferred to an excel file as required by the journal formatting guide. This file also contains the list of mRNAs with a large oligo-adenylated fraction.

- The scatterplot showing the relationship between mean poly(A)-tail length and the % recovery of mRNAs in the poly(A)+ fraction from the total sample has been revised to adhere to the journal formatting requirements. The revised Supplementary Figure 1j shows both the mean of 2 to 4 wild-type replicates along with values used to obtain it. Furthermore, additional data points specify the mean poly(A)-tail length calculated from the total fraction DRS sequencing for comparison.

- To improve the story flow, the information regarding XUT and SUT metabolism in *pap1-1* background has been removed from the Introduction and Results sections, as it provided limited added value. SUTs and XUTs are non-coding RNAs with no assigned function and metabolism similar to mRNAs. Sentences and Figures removed are specified below:

'Pap1-dependent ncRNA SUTs and XUTs had initial poly(A)-tail lengths of 44 and 48 adenosines and were shortened by ~10 and ~14 adenosines in pap1-1. The loss of RRP6, but not TRF4, stabilized these species about two-fold, without significantly rescuing poly(A)-tail length (Supplemental Fig. 2h).'

'SUTs and XUTs are CPF-dependent and presumably mainly cytoplasmic ncRNAs. As expected, they had poly(A)-tails of around 40 adenosines, which were insensitive to the exosome, while the bulk of those ncRNAs was upregulated in rrp6Δ and trf4Δ cells, as previously described11 (Supplemental Fig. 3d).'

'SUTs and XUTs follow a similar pattern to mRNAs and attain poly(A)-tail lengths of 58 and 60 adenosines in double-depleted cells (Supplemental Fig. 4g).'

h
Removed Supplemental Figure 2h, The top panel shows a violin plot of mean poly(A)-tail lengths of SUTs and XUTs in WT, *pap1-1*, *pap1-1 rrp6Δ*, and *pap1-1 trf4Δ* cells. The bottom panel shows the sum of all DRS counts mapped to those ncRNA loci. Because of a large decrease in mRNA levels noted in *pap1-1* cells, the sum of ncRNA counts is shown as raw counts, in contrast to similar graphs presented in the subsequent figures, normalized to mRNA levels.

d
Removed Supplemental Figure 3d, The top panel shows a violin plot of mean poly(A)-tail lengths of SUTs and XUTs in WT, *rrp6Δ*, *trf4Δ*, *trf5Δ*, and *rrp6Δ trf4Δ* cells. The bottom panel shows the sum of all DRS counts mapped to those ncRNA loci, normalized to the sum of mRNA counts.

Removed Supplemental Figure 4g, The top panel shows a violin plot of mean poly(A)-tail lengths of SUTs and XUTs in WT, *DIS3-AA*, *DIS3-RRP6-AA*, and *ski2Δ* cells. The bottom panel shows the sum of all DRS counts mapped to those ncRNA loci, normalized to mRNA counts.

- Supplemental Figure 2g was removed. This panel showed the % of mRNA recovery in the poly(A)+ fraction from the total by RT-qPCR (polyA+/total mRNA levels). The raw data is still displayed in Supplemental Figure 2b, thus removal of this panel does not result in information loss.

- in the main Fig. 2h, violin plots for *SCW11* mRNA were replaced with *TDH1*. *SCW11* was moved to Supplemental Fig. 2f. As a consequence due to space limitation one of the examples from the Supplemental Fig. 2f was removed.

- As a consequence of removing all information related to *pop2Δ*, as suggested by Reviewer 2, the Figure 6c, showing the susceptibility of mRNAs up- and down-regulated following heat-stress to exonucleolysis by Ccr4, Pan2 or the exosome has been updated to exclude mRNAs targeted by Pop2 and is now presented in a form of pie charts.

- The Materials and Methods chapter was up-dated with a section describing production of spike-in

for DRS sequencing and analysis of the total DRS samples. The latter was detailed in response to Reviewers 1 and 2. In addition, the sections describing the bioinformatic analysis was divided into separate sub-sections to provide more clarity to the reader, and a separate chapter describing the source of previously published data was added.

REVIEWERS' COMMENTS

Reviewer #1 (Remarks to the Author):

The authors responded very convincingly to my suggestions/questions and did so by adding very informative additional data (in the supplemental data).

As far as I am concerned, I think that this manuscript should now be published as it is.

Sincerely,

Alain Jacquier

Reviewer #2 (Remarks to the Author):

This is a very thorough and comprehensive revision that substantially strengthened and improved the manuscript. I thank the authors for their careful and diligent approach toward addressing my concerns and critiques. This study will generate interest in the field and I recommend its publication.

Reviewer #3 (Remarks to the Author):

All previous issues and comments have been addressed. A few minor comments to the current revision:

Line 143: Change "transcription level" to "mRNA synthesis rate"

Figure 1f: Re-scale X axis on upper left graph to match graphs below.

Line 161: Current data demonstrates the point sufficiently, but I am curious whether the authors have tried nuclear isolation followed by RNA extraction? That approach might reduce noise from cytoplasmic RNA.

Figure 2a: Library counts are consistently smaller in 38°C MEX67-AA cells compared to 38°C Cells. Could that be an effect of inefficient lysis of nuclei during RNA extraction?

Line 163: Remove 37°C from this sentence, as it is not relevant to figure 2.

Figure 3c: For completeness, transcripts marked in green *rrp6D/WT* and *trf4D rrp6D/rrp6D*, should also be marked in green for *trf4D/WT* and *trf5/WT*.

Figure 7: The data supports the majority of the model presented here. However, I am not entirely convinced about the importance of poly-A elongation by *trf4*. The data only demonstrate this for *rrp6* sensitive targets (Figure 3c). I would be convinced if the green labelled targets in figure 3c also respond in *trf4/WT* cells. If *trf4* only elongate poly-A in *rrp6D* cells, I would question the biological relevance.

We wish to thank the Reviewers and the Editors for the insightful remarks which helped us to improve our work. Please find specific responses to the Reviewers comments below - other modifications requested by Editors are listed in a separate letter.

REVIEWERS' COMMENTS

Reviewer #1 (Remarks to the Author):

The authors responded very convincingly to my suggestions/questions and did so by adding very informative additional data (in the supplemental data).

As far as I am concerned, I think that this manuscript should now be published as it is.

Sincerely,

Alain Jacquier

Reviewer #2 (Remarks to the Author):

This is a very thorough and comprehensive revision that substantially strengthened and improved the manuscript. I thank the authors for their careful and diligent approach toward addressing my concerns and critiques. This study will generate interest in the field and I recommend its publication.

Reviewer #3 (Remarks to the Author):

All previous issues and comments have been addressed. A few minor comments to the current revision:

Line 143: Change “transcription level” to “mRNA synthesis rate”

This was corrected.

Figure 1f: Re-scale X axis on upper left graph to match graphs below.

This was corrected.

Line 161: Current data demonstrates the point sufficiently, but I am curious whether the authors have tried nuclear isolation followed by RNA extraction? That approach might reduce noise from cytoplasmic RNA.

We have considered such experiments. Sadly, however, we do not have in our hands an efficient protocol for the isolation of yeast nuclei. The yeast cell is the size of a mammalian nucleus, and thus published protocols for mammalian cell lines cannot be simply applied to yeast samples. Nevertheless, this is surely an interesting experiment to try in the mammalian system and we thank the reviewer for this suggestion.

Figure 2a: Library counts are consistently smaller in 38°C MEX67-AA cells compared to 38°C Cells. Could that be an effect of inefficient lysis of nuclei during RNA extraction?

All RNA samples were prepared using the hot acid phenol method (cells are resuspended in an aqueous buffer containing SDS and mixed with phenol pH 4.3 – for details, see materials and methods section), which should efficiently break cells regardless of their growth conditions. The low count in those libraries is rather due to poor sequencing depth for those samples. The nanopore technology is strongly depends on the quality of the sample and of the sequencing flowcell itself. Since all samples were sequenced as spike-ins for mammalian or plant RNA samples, it is difficult to determine if the

low count is due to an accompanying contaminant in the yeast, plant, mammalian RNA sample or poor quality of the flowcell itself.

Line 163: Remove 37°C from this sentence, as it is not relevant to figure 2.

This was corrected.

Figure 3c: For completeness, transcripts marked in green *rrp6D*/WT and *trf4D rrp6D/rrp6D*, should also be marked in green for *trf4D*/WT and *trf5*/WT.

Figure 7: The data supports the majority of the model presented here. However, I am not entirely convinced about the importance of poly-A elongation by *trf4*. The data only demonstrate this for *rrp6* sensitive targets (Figure 3c). I would be convinced if the green labelled targets in figure 3c also respond in *trf4*/WT cells. If *trf4* only elongate poly-A in *rrp6D* cells, I would question the biological relevance.

Joint response to comment to Figure 3c and Figure 7:

As mentioned in the original text, the *trf4Δ* strain does not display a substantial elongation or shortening of poly(A)-tails (mean change of mRNA poly(A)-tails was 0.5 As). The Trf4 poly(A)-polymerase activity directed towards mRNAs is only clearly visible in Pap1-impaired cells and in *rrp6Δ* as demonstrated by relevant sections of the manuscript. The revised scatterplot shows that mRNAs with poly(A)-tails elongated by at least 10 adenosines in *rrp6Δ* are largely unchanged in length in *trf4Δ*. Those transcripts have the tendency to be elongated in *trf5Δ*, though not more than the general population. Collectively this shows that mRNA poly(A)-tails are mostly substrates for exonucleolysis by the exosome subunits Dis3 and Rrp6, which are assisted by Trf5 in a way not involving its poly(A)-polymerase activity. In turn, the Trf4 poly(A)-polymerase activity is only evidenced in conditions of Rrp6 impairment and only affects a limited number of transcripts. We acknowledge that for wild-type cells Trf4 is irrelevant for mRNA poly(A)-tail metabolism. However, it is known that in wild-type yeast cells, Rrp6 protein is down-regulated during meiosis (Lardenois et al., 2011). Thus, it is not to be excluded that Trf4 activity could regulate poly(A)-tail biogenesis of some specific mRNAs during meiosis or some other yet unidentified growth stage.

We understand why the model presented in Figure 7 was misleading. We have thus put the Trf4-elongation part in a smaller font and grey color to decrease its apparent significance.

Lardenois A, Liu Y, Walther T, Chalmel F, Evrard B, Granovskaia M, Chu A, Davis RW, Steinmetz LM, Primig M. Execution of the meiotic noncoding RNA expression program and the onset of gametogenesis in yeast require the conserved exosome subunit Rrp6. Proc Natl Acad Sci U S A. 2011 Jan 18;108(3):1058-63. doi: 10.1073/pnas.1016459108. Epub 2010 Dec 13. PMID: 21149693; PMCID: PMC3024698.

OTHER MANUSCRIPT MODIFICATIONS

As requested by the Editors, Supplemental Tables 1 and 5 are now Supplemental Data 1 and Supplemental Data 2, respectively. The numbering of the remaining Supplemental Tables changed accordingly.

The key aggregating RNA fractions (poly(+), poly(A)-, or CUT) was added to Supplemental Fig. 1a, Supplemental Fig 2b, and Supplemental Figure 2d.

A detailed protocol for isolation of poly(A)⁺ fraction suggested by the manufacturer of the Dynabeads oligo(dT)₂₅ kit was also included in the Methods section.